# The Lighthouse of Language: Enhancing LLM Agents via Critique-Guided Improvement

**Ruihan Yang**[1,♡,∗] **Fanghua Ye**[♠,†] **Jian Li**[♠,†] **Siyu Yuan**[1,♡], **Yikai Zhang**[2,♡],
**Zhaopeng Tu**[♠], **Xiaolong Li**[♠], **Deqing Yang**[1,♡†]

[1,♡]School of Data Science, Fudan University    [♠]Tencent Hunyuan

[2,♡]College of Computer Science and Artifcial Intelligence, Fudan University

[♡]{rhyang17, yangdeqing}@fudan.edu.cn    [♡]{syyuan21, ykzhang22}@m.fudan.edu.cn

[♠]{fanghua.ye.21, lijianjack, tuzhaopeng}@gmail.com

Project Page: https://github.com/rhyang2021/CGI

## Abstract

Large language models (LLMs) have recently transformed from text-based assistants to autonomous agents capable of planning, reasoning, and iteratively improving their actions. While numerical reward signals and verifiers can effectively rank candidate actions, they often provide limited contextual guidance. In contrast, natural language feedback better aligns with the generative capabilities of LLMs, providing richer and more actionable suggestions. However, parsing and implementing this feedback effectively can be challenging for LLM-based agents. In this work, we introduce **C**ritique-**G**uided **I**mprovement (CGI), a novel two-player framework, comprising an actor model that explores an environment and a critic model that generates detailed nature language feedback. By training the critic to produce fine-grained assessments and actionable revisions, and the actor to utilize these critiques, our approach promotes more robust exploration of alternative strategies. Experiments in three interactive environments show that CGI outperforms existing baselines by a substantial margin. Notably, even a small critic model surpasses GPT-4 in feedback quality. The resulting actor achieves state-of-the-art performance, demonstrating the power of explicit guidance to enhance decision-making in LLM-based agents.

## 1 Introduction

Recent advances in large language models (LLMs) [1; 2; 3] have broadened their applicability from conventional chat and text-based interactions to more dynamic, agentic roles. In these roles, LLM-based agents [4; 5; 6] harness the reasoning and planning capabilities of LLMs to automate complex tasks across diverse domains, including code generation [7; 8], software engineering [9; 10], and web applications [11; 12]. A key requirement in these agentic tasks is the ability to iteratively acquire, store, and use new information to improve performance. Thus, a central challenge emerges: *How can high-quality feedback be obtained and utilized efficiently?*

A common approach for providing feedback is to rely on numerical signals, such as verifiers [13; 14] or reward models [15; 16]. In this paradigm, LLMs are typically trained as discriminators to evaluate and select the best action from a set of candidates. One widely used method is Best-of-$N$ (BoN) [17; 18], where the agent generates $N$ possible actions and a trained reward model picks the

---

[∗]Work done during an internship at Tencent Hunyuan.

[†]Corresponding authors.

39th Conference on Neural Information Processing Systems (NeurIPS 2025).

highest-scoring one. However, numerical feedback has limited informativeness: it reduces guidance to scalar scores that merely filter candidate actions, overlooking context-specific suggestions, avenues for exploration, or strategies for refining suboptimal behavior.

A more direct and flexible approach involves using natural language feedback. Rather than providing only a "best" action, LLMs can generate detailed critiques and explicit, context-aware recommendations for improvement. However, transitioning to natural language feedback presents two main challenges: *1) Weak Feedback*: Many techniques rely on self-refinement or self-correction [19; 20; 21], which heavily depend on the LLM's inherent capacity. This reliance can lead to degraded performance, especially when models hallucinate or encounter complex problems [20; 22]; *2) Poor Utilization*: Unlike numerical feedback, which directly selects an action to execute, verbal critiques require the agent to accurately interpret and integrate the guidance into its decision-making process. LLM agents often struggle with this, sometimes demonstrating limited flexibility in adjusting or discarding previously generated plans [23; 24].

In response, we propose **C**ritique-**G**uided **I**mprovement (CGI), a two-player framework in which an actor model interacts with the environment while a critic model provides supervisory feedback. CGI features two main stages: **Critique Generation** and **Action Refinement**. In the Critique Generation stage, we train a critic model to produce precise evaluations and actionable revision instructions. In the Action Refinement stage, the actor model learns to effectively apply these critiques through iterative supervised fine-tuning, thereby improving both its reasoning and its ability to integrate external feedback.

We conduct extensive experiments in three interactive environments [11; 25; 26], demonstrating that CGI substantially enhances performance over baseline approaches. Notably, our critic model, when trained on a small backbone (e.g., Llama-3-8B), outperforms even GPT-4 as a critic by a large margin (+29.16%). Additionally, the action-refinement process further boosts results, surpassing state-of-the-art metrics by 26.74%.

In summary, our contributions are as follows: *1)* We tackle the challenge of obtaining and utilizing high-quality nature language feedback in agentic tasks. *2)* We present Critique-Guided Improvement (CGI), a novel two-player framework that provides more informative feedback to LLM-based agents by combining a specialized critic model with an actor model. *3)* We develop a training strategy that iteratively refines the actor's actions based on natural language critiques, enabling both improved reasoning and enhanced adaptability. *4)* We conduct extensive experiments across three diverse interactive environments. By leveraging CGI, the agent can continuously improve its performance in long-horizon tasks and enhance task efficiency by achieving higher scores in fewer steps.

## 2   Related Work

**Learning from Feedback**   Current feedback can be categorized into numerical and natural language feedback. Numerical feedback [27; 28; 29; 30] is typically provided by training a reward model (RM) or verifiers. In this approach, the model learns to predict either the correctness of a solution [13; 29] or the preference between multiple solutions [17; 31]. Specifically, the RM generates a continuous numerical score, which is then incorporated into a classification objective. In contrast, natural language feedback offers denser rewards by evaluating the model's actions using natural language. This feedback can be generated in two main ways: by prompting an off-the-shelf LLM to act as a verifier (*i.e.*, LLM-as-judge) [32; 33], or through self-refinement and self-critique [20; 34], where the model evaluates and improves its own outputs. Our trained critic model provides natural language feedback, offering denser and more accurate rewards.

**Agent Learning in Interactive Environments.**   Previous approaches to agent learning in interactive environments can be classified into three main categories: *1)* Prompt-based methods [21; 35; 36; 37] utilize human-written prompts to guide LLMs in summarizing experiences. These summaries, which may include causal abstractions from both successful and failed attempts [20; 38] or transferable skills [39], are integrated into the model's memory to enhance its knowledge and performance. *2)* Training-based methods [40; 12; 41] rely on techniques such as Supervised Fine-Tuning (SFT) [42; 43] or Direct Preference Optimization (DPO) [44; 45]. to train LLMs. The training data may come from expert models or be generated through exploration strategies like Monte Carlo Tree Search (MCTS). *3)* Inference-time sampling methods [14; 15; 16] employ techniques such as Best-of-N

(BoN) [17; 18] and Tree-of-Thought (ToT) [46] to identify optimal actions during inference. These methods leverage the prior knowledge in LLMs, enabling more efficient search processes. Our CGI method introduces a novel variation of inference-time sampling by using a trained critic that evaluates candidate actions and suggests revisions to enhance inference performance.

# 3 Preliminary

**Partially Observable Markov Decision Process** We define the collection of environments as $\mathcal{E}$. For a specific environment $e \in \mathcal{E}$, the embodied tasks for LLM agents are typically modeled as a Partially Observable Markov Decision Process (POMDP): $(\mathcal{X}, \mathcal{S}, \mathcal{A}, \mathcal{O}, \mathcal{T})_e$. Here, $\mathcal{X}$ denotes the set of instructions, $\mathcal{S}$ represents the set of environment states, $\mathcal{A}$ is the set of available actions at each state, and $\mathcal{O}$ represents the observations available to the agent. The transition function $\mathcal{T} : \mathcal{S} \times \mathcal{A} \rightarrow \mathcal{S}$ is defined by the environment, while the reward function $\mathcal{R} : \mathcal{S} \times \mathcal{A} \rightarrow [0, 1]$ specifies the reward received by the agent from the environment. For an actor model $\pi_\theta$ parameterized by $\theta$, the actor selects the next action $a_t \sim \pi_\theta(a|\tau_t, e)$ at each time step $t$, based on the interaction history $\tau_t$, which is defined as
$$\tau_t = \{x, a_0, o_0, \ldots, a_{t-1}, o_{t-1}\}, \quad \tau_0 = \{x\}.$$
The trajectory is then represented as
$$\tau = (x, a_0, o_0, \ldots, a_T, o_T) \sim \pi_\theta(\tau \mid x, e).$$

**Iterative Supervised Fine-Tuning in Agentic Task** Iterative Supervised Fine-Tuning (SFT) is a process of exploration and learning [47; 48; 49]. It iteratively utilizes the correct responses from the actor model to enhance the model's problem-solving abilities. The process involves $K$ iterations, each consisting of two steps: exploration and learning. For the collection of environments $\mathcal{E}$, in the exploration step of iteration $k$, the model $\pi_\theta^{k-1}$ from the previous iteration is applied to each environment $e$, resulting in $|\mathcal{E}|$ trajectories $\{\tau^{(j)}\}_{j=1}^{|\mathcal{E}|}$. These trajectories are then filtered using the reward function $\mathcal{R}(\tau)$, retaining only the correct ones. The filtered trajectories form a new dataset $\mathcal{D}_{\text{correct}} = \{\tau^{(j)}\}_{j=1}^{N'}$, where $N'$ is the number of trajectories retained. In the learning step of iteration $k$, this new dataset is used to fine-tune the actor model $\pi_\theta$ to obtain $\pi_\theta^k$.

# 4 Methodology

In this section, we first provide an overview of the proposed Critique-Guided Improvement (CGI) framework (§ 4.1), explaining how the actor and critic collaborate to enhance performance. We then detail the two key stages of CGI, critique generation and action refinement, which respectively address the challenges of critic's *weak feedback* and the actor's *poor utilization*. In the **Critique Generation** stage (§ 4.2), the critic model learns to evaluate the actor's candidate actions and provide actionable feedback. In the **Action Refinement** stage (§ 4.3), the actor refines its actions by integrating critiques from the critic model into its interaction with the environment. Algorithm 1 summarizes the CGI framework (see Appendix C for definitions of all notifications).

## 4.1 Overview of the CGI Framework

The CGI framework follows a two-player setting, where the actor model $\pi_\theta$ generates multiple candidate actions, and the critic model $\pi_\phi$ provides feedback to refine them. For a specific environment $e \in \mathcal{E}$, at each time step $t$, given the refined trajectory history
$$\tau_t' = \{x, a_0', o_0, \ldots, a_{t-1}', o_{t-1}\},$$
the actor generates $M$ candidate actions, stored in the action buffer $A_t = \{a_{t,i}\}_{i=1}^M$. The critic then evaluates each candidate and generates corresponding critiques $c_t = \{\pi_\phi(c \mid \tau_t', a_{t,i}, e)\}_{i=1}^M$. Using these critiques, the actor refines its decision and generates the final action $a_t' = \pi_\theta(a \mid \tau_t', c_t, e)$, which is then executed in the environment. This iterative process produces the refined trajectory:
$$\tau' = \{x, a_0', o_0, \ldots, a_T', o_T\}.$$
In this collaborative framework, higher-quality critiques $c_t$ and better utilization of feedback lead to improved refined actions $a_t'$, ultimately enhancing the final performance $\mathcal{R}(\tau')$.

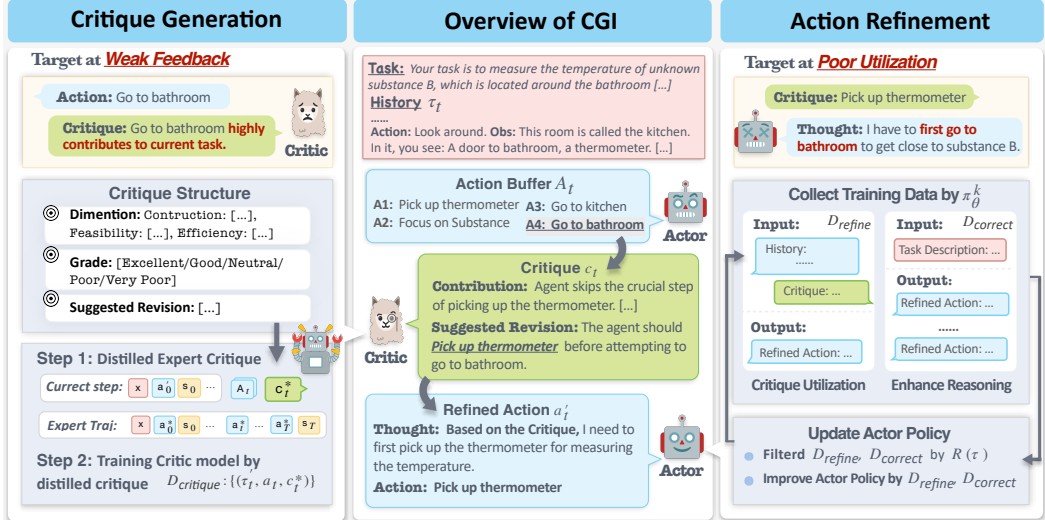

Figure 1: An overview of CGI, illustrated with a ScienceWorld example. The actor first generates candidate actions, and the critic then assesses each action and generates suggested revisions. The actor refines its actions based on the critique. The **Critique Generation** stage enhances the critic model's ability to provide effective feedback, while the **Action Refinement** stage helps the actor model better utilize the critique.

## 4.2   Critique Generation

The **Critique Generation** stage trains the critic model to assess the alignment between LLM-generated actions and optimal ones at each step, and convert this assessment into structured natural language critiques. We first define the critique structure, then outline the training method for critic.

**Critique Structure**   The critique generated by the critic model $\pi_\phi$ consists of two components: *Discrimination*, which involves analyzing and assessing the quality of the candidate action, and *Revision*, which provides actionable suggestions for improvement.

For the discrimination component, the critic model evaluates the candidate action in three predefined dimensions:

- `Contribution`: Assessing how well the candidate action contributes to solving the overall task. Irrelevant actions receive a low score.
- `Feasibility`: Determining whether the candidate action is valid according to the agent's predefined action list.
- `Efficiency`: Evaluating whether the candidate action approaches the task optimally, avoiding unnecessary steps or redundancy.

For the revision component, the critic model first assigns an overall grade to each candidate action (*i.e.*, `[Excellent/Good/Neutral/Poor/Very Poor]`), then generates concise and actionable suggestions based on its analysis of the candidate actions (see Appendix D for detailed prompts).

**Fine-tuning the Critic Model**   To collect high-quality step-level expert critiques, we employ an expert critic annotator (e.g., GPT-4 [1]) to generate critiques for each candidate action based on a reference expert trajectory. For a specific environment $e \in \mathcal{E}$, at time step $t$, the actor generates candidate actions $a_t$. Given the refined history trajectory $\tau'_t$ and the expert trajectory $\tau^{\mathrm{exp}}$, the expert critic $\pi_{\mathrm{exp}}$ assesses the alignment between candidate actions and the optimal ones, converting this assessment into structured critiques according to a predefined format. Formally, the critiques for each candidate action are represented as: $c_t \sim \pi_{\mathrm{exp}}(c \mid a_t, \tau'_t, \tau^{\mathrm{exp}}, e)$.

Subsequently, the actor refines its next step by generating $a'_t$ based on the critique $c_t$. Critiques are collected only when $\mathcal{R}(\tau') = 1$, ensuring they originate from correct trajectories. We collect step-level

---

**Algorithm 1** Critique-Guided Improvement (CGI)

---

1: **Input:** Initialized policy actor model $\pi_\theta$, critic model $\pi_\phi$, environment set $\mathcal{E}$, full instruction set $\mathcal{X}$, reward function $\mathcal{R}$, the expert critiques $\mathcal{D}_{\text{critique}}$, the expert path $\mathcal{D}_{\text{expert}}$, the general dataset $\mathcal{D}_{\text{general}}$, max time step $T$, and the iteration rounds $K$.

2: **Procedure Fine-tune the Critic Model:**

3:     Minimize the following objective to obtain the critic model $\pi_\phi$:
$$\mathcal{L}_{\text{critic}}(\phi) = \mathbb{E}_{(c_t, \tau'_t, a_t, e) \sim \mathcal{D}_{\text{critique}}} \Big[ \log \pi_\phi \left( c_t \mid \tau'_t, a_t, e \right) \Big];$$

4: **Procedure Iterative Action Refinement:**

5:     $\pi_\theta^0 \leftarrow \pi_{\theta_{base}}$;

6: **for** iteration $k = 1$ to $K$ **do**

7:     $D_{\text{train}} \leftarrow D_{\text{expert}}$; // Initialize $\mathcal{D}_{\text{train}}$ with $D_{\text{expert}}$.

8:     **Perform** Exploration Step
        Actor $\pi_\theta^{k-1}$ interacts with $e \in \mathcal{E}$ under the guidance of critic $\pi_\phi$;

9:        // Collect critique-action pairs with $\mathcal{R}(\tau') = 1$.

10:        Collect $D_{\text{refine}} = \bigcup_{e \in \mathcal{E}} D_{\text{refine}}^e$, where $D_{\text{refine}}^e = \{(\tau'_t, c_t, a'_t)\}_{t=1}^T$;

11:        // Update training set with correct trajectory.

12:        Collect $D_{\text{correct}} = \bigcup_{e \in \mathcal{E}} D_{\text{correct}}^e$, $D_{\text{train}} \leftarrow D_{\text{train}} \cup \mathcal{D}_{\text{correct}}$;

13:     **Perform** Learning Step

14:        Minimize the following objective to obtain actor model $\pi_\theta^k$:
$$\mathcal{L}_{\text{actor}}(\theta) = \beta \left\{ \mathbb{E}_{(\tau, x, e) \sim \mathcal{D}_{\text{train}}} \Big[ \log \pi_\theta(\tau \mid x, e) \Big] + \mathbb{E}_{(a'_t, \tau'_t, c_t, e) \sim \mathcal{D}_{\text{refine}}} \Big[ \log \pi_\theta(a'_t \mid \tau'_t, c_t, e) \Big] \right\}$$
$$+ (1 - \beta) \, \mathbb{E}_{(x, y) \sim \mathcal{D}_{\text{general}}} \Big[ \log \pi_\theta(y \mid x) \Big];$$

15: **end for**

---

expert critiques from each environment, forming the dataset $\mathcal{D}_{\text{critique}} = \cup_{e \in \mathcal{E}} \mathcal{D}_{\text{critique}}^e$ . The critic model is then fine-tuned using supervised learning with the collected expert critiques. Specifically, we apply the standard language modeling loss, defined as:

$$\mathcal{L}_{\text{critic}}(\phi) = \mathbb{E}_{(c_t, \tau'_t, a_t, e) \sim \mathcal{D}_{\text{critique}}} \Big[ \log \pi_\phi \left( c_t \mid \tau'_t, a_t, e \right) \Big].$$

This approach enables the critic model to generate structured, step-level critiques that provide both discrimination and revision, which are crucial for guiding the actor model toward more effective decision-making.

### 4.3 Action Refinement

Although the critic model is trained to provide high-quality critiques, the actor may not fully utilize the feedback. Therefore, action refinement is necessary to enhance the actor's ability to effectively leverage critiques for improving its actions. A key challenge in this process is policy misalignment. During training, the actor model learns to incorporate critiques based on its current policy. However, after training, the model's policy may evolve, making it difficult to integrate critiques effectively for newly generated candidate actions. This misalignment can lead to suboptimal action refinement.

To address this issue, we propose an iterative action refinement method based on supervised fine-tuning (SFT). This method consists of two main components: exploration and learning. In the exploration step at iteration $k$, for a specific environment $e \in \mathcal{E}$, the actor model $\pi_\theta^{k-1}$ interacts with the environment under the guidance of the critic model $\pi_\phi$. At each time step $t$, the refined action is given by $a'_t = \pi_\theta(a \mid \tau'_t, c_t, e)$, forming a set of critique-action pairs $\{(\tau'_t, c_t, a'_t)\}_{t=1}^T$. Here, $\tau'_t$ represents the previously refined trajectory, and $c_t$ denotes the current critique. To ensure the quality of the trajectories, we filter them based on the environment reward $\mathcal{R}$. Only trajectories for which $\mathcal{R}(\tau') = 1$, along with their corresponding critique-action pairs, are retained. We collect correct trajectories and critique-action pairs from each environment, yielding two datasets: $\mathcal{D}_{\text{correct}} = \cup_{e \in \mathcal{E}} \mathcal{D}_{\text{correct}}^e$ and $\mathcal{D}_{\text{refine}} = \cup_{e \in \mathcal{E}} \mathcal{D}_{\text{refine}}^e$. The dataset $\mathcal{D}_{\text{correct}}$ enhances the model's reasoning ability by providing correct trajectories, while $\mathcal{D}_{\text{refine}}$ improves its capacity to utilize critiques and generate refined actions.

In the learning step of iteration $k$, the newly collected datasets, combined with the expert path $\mathcal{D}_{\text{expert}}$ for each environment, are used to fine-tune the actor model $\pi_\theta$. To avoid overfitting, we follow previous work [50] by fine-tuning the original model $\pi_\theta$ rather than the previous iteration model $\pi_\theta^{k-1}$. Furthermore, following AgentTuning [51], we incorporate general datasets such as ShareGPT3[3] to improve generalization. The training objective is defined as:

$$\mathcal{L}_{\text{actor}}(\theta) = \beta \left\{ \mathbb{E}_{(\tau, x, e) \sim \mathcal{D}_{\text{train}}} \left[ \log \pi_\theta(\tau \mid x, e) \right] + \mathbb{E}_{(a_t', \tau_t', c_t, e) \sim \mathcal{D}_{\text{refine}}} \left[ \log \pi_\theta(a_t' \mid \tau_t', c_t, e) \right] \right\}$$
$$+ (1 - \beta) \, \mathbb{E}_{(x, y) \sim \mathcal{D}_{\text{general}}} \left[ \log \pi_\theta(y \mid x) \right],$$

where $\mathcal{D}_{\text{train}} = \mathcal{D}_{\text{expert}} \cup \mathcal{D}_{\text{correct}}$. After this refinement step, a new dataset with higher-quality samples is generated for further training. Overall, this iterative action refinement process allows the actor model to progressively improve its reasoning capabilities and better integrate critiques through continuous interaction with the environment.

## 5 Experiment Settings

In this section, we conduct extensive experiments in three interactive environments to demonstrate the effectiveness of the critic model and our CGI framework.

### 5.1 Interactive and Agentic Environments

Following previous work [40; 52], we conduct experiments on three types of representative interactive environments:

- WebShop [11], which is an interactive web environment for online shopping. It contains 12K instructions and offers over one million real products from amazon.com. Agents can click buttons on the webpage or perform searches using the search engine.

- ScienceWorld [25], which is a text-based scientific environment designed to evaluate agents' scientific reasoning abilities. It includes 30 types of scientific tasks at the standard elementary science curriculum level.

- TextCraft [26], which is a text-based environment to create Minecraft items. It constructs a crafting tree based on Minecraft's recipes. Each task provides a target item and a list of crafting commands generated by the tree. Agents receive a reward of 1 when they successfully craft the target item.

**Evaluation Metrics**   Following the setup of AgentGym [40], we evaluate our model on the test sets for these three environments (200 simulations for ScienceWorld and WebShop, 100 for TextCraft). For ScienceWorld and WebShop, we use the average final score as the evaluation metric. For TextCraft, we use the success rate as the evaluation metric. Further details can be found in Appendix E.

### 5.2 Training Settings

We use Llama-3-8B-Instruct [3] as the backbone model for both the actor and critic models. To collect training data, we randomly sample 500 simulations from WebShop, 350 from ScienceWorld, and 374 from TextCraft. We train the critic model by using the expert critic (*i.e.*, GPT-4o) to guide the actor in interacting with the environment three times, collecting expert critiques during each interaction. For action refinement, we perform three iterations and report the results of the third iteration in Table 2. The training data for the critic model and each iteration of action refinement are provided in Table 4. Additional details can be found in Appendix F.

### 5.3 Baselines

To evaluate the effectiveness of our critic model, we fix the actor model as Llama-3-8B-Instruct [3]. We then compare our critic model against two types of approaches (see Appendix G for implementation details): *1)* **Numerical based**: We use DGAP [14], a discriminator trained to assess the alignment between actor actions and expert actions at the step level, and Explicit RM [15], which

---

[3]https://huggingface.co/datasets/Vtuber-plan/sharegpt-cleaned

Table 1: We compared the natural language feedback provided by the trained critic model with other methods, including numerical-based (*i.e.*, DGAP, Explict RM), and verbal-based (*i.e.*, self-critique, GPT-4o). Our critic offers better guidance to the actor model in all three interactive scenarios.

| Model | Method | WebShop | ScienceWorld | TextCraft | Average |
|---|---|---|---|---|---|
| *Llama-3-8B-Instruct* | No Critique | 13.49 | 14.48 | 10.00 | 12.65 |
| | DGAP | 30.41 | 19.52 | 21.00 | 23.64 |
| | Explicit RM | 14.21 | 18.41 | 15.00 | 15.87 |
| | Self-Critique | 1.50 | 10.06 | 19.00 | 10.19 |
| | GPT-4o | 17.78 | 33.06 | 46.00 | 32.28 |
| | Critic Model (*Ours*) | **56.80** | **68.51** | **59.00** | **61.44** |
| *Llama-3-70B-Instruct* | No Critique | 8.35 | 49.20 | 2.00 | 19.85 |
| | DGAP | 11.17 | 55.86 | 9.00 | 25.34 |
| | Explicit RM | 11.18 | 48.41 | 14.00 | 24.53 |
| | Self-Critique | 3.00 | 28.17 | 40.00 | 23.72 |
| | GPT-4o | 16.24 | 43.73 | 56.00 | 38.65 |
| | Critic Model (*Ours*) | **52.20** | **72.44** | **73.00** | **65.88** |
| *Llama-3-8B-Instruct + SFT* | No Critique | **76.12** | 32.85 | 46.00 | 51.66 |
| | DGAP | 73.97 | 38.26 | 56.00 | 56.08 |
| | Explicit RM | 74.33 | 41.57 | 52.00 | 55.97 |
| | Self-Critique | 50.18 | 31.22 | 21.00 | 34.13 |
| | GPT-4o | 55.65 | 48.48 | **58.00** | 54.04 |
| | Critic Model (*Ours*) | 74.68 | **55.94** | 56.00 | **62.21** |

is trained to predict the Q-value, *i.e.*, expected accumulated rewards at each time step. *2)* **Verbal based**: We employ a self-critique method where the actor model itself generates critiques for each candidate action at the step level. Additionally, we use GPT-4o (gpt-4o-2024-08-06) [1] as a critic, which serves as a strong general-purpose evaluator. The structure of the critiques aligns with that in Appendix D. Each approach guides the actor's inference, and we evaluate their effectiveness based on the actor's performance[4].

For the assessment of CGI, consistent with the methodology used in AgentGym [40], we select a combination of closed-source models, including GPT-3.5-turbo [53], GPT-4o [54], Claude 3 [55], and DeepSeek-Chat [56], as well as open-source models such as Llama-3-70B-Instruct [3]. Additionally, we include agents trained on expert trajectories, such as AgentLM (13B and 70B) [57] and Agent-Flan [42]. We also compare our approach to Iterative SFT, which iteratively refines the actor model using correct trajectories collected from interactions with the environment, and Reflexion [20], a self-reinforcement method that concludes each iteration with a summary to guide decision-making in subsequent iterations. We report the results from the third iteration for both approaches.

## 6 Main Results

We first evaluate our critic model against numerical- and verbal-based methods to assess its effectiveness. As shown in Table 1, our 8B critic model significantly outperforms GPT-4o on both the Llama-3 (8B and 70B) and Llama-3-8B's fine-tuned variant trained on expert data. Furthermore, Table 2 demonstrates that through iterative action refinement of the actor model, our CGI notably enhances actor performance in interactive environments. It outperforms both advanced closed-source models (e.g., GPT-4o) and agents trained on expert trajectories (e.g., AgentLM-70B and Agent-FLAN). Based on these results, we identify three key findings.

**Finding 1: Verbal critique feedback is more effective than numerical signal.** As shown in Table 1, the Critic Model consistently outperforms numerical feedback from the discriminator. For the Llama-3-8B model, it achieves an average improvement of 42.89% over the no-critique baseline, while the discriminator yields only a 5.09% gain. The advantage persists with the stronger Llama-3-70B and the fine-tuned Llama-3-8B models, where the Critic Model delivers a 46.03% and 10.55% improvement, respectively, compared to 5.49% and 4.42% from the discriminator. These results indicate that the discriminator's numerical feedback relies heavily on the model's inherent capabilities, as it essentially performs action-level filtering. For example, in the ScienceWorld, the discriminator enables Llama-3-70B to reach 55.86%, but only 18.52% for Llama-3-8B. In contrast, the Critic

---

[4]In the subsequent experiments of this paper, we set the number of candidate actions sampled at inference time to $M = 5$.

Table 2: Results from three interactive environments. The CGI here employs a two-player setting with a fine-tuned LLama-3-8B critic model and an 8B actor model refined through three iterations. We compare its performance with various models.

| Method | WebShop | ScienceWorld | TextCraft | Average |
|---|---|---|---|---|
| *Closed-source Models* | | | | |
| DeepSeek-Chat | 11.00 | 16.80 | 23.00 | 16.93 |
| Claude-3-Haiku | 5.50 | 0.83 | 0.00 | 2.11 |
| Claude-3-Sonnet | 1.50 | 2.78 | 38.00 | 14.09 |
| GPT-3.5-Turbo | 12.50 | 7.64 | 47.00 | 22.38 |
| GPT-4o | 25.48 | 46.91 | 64.00 | 45.46 |
| *Open-source Models* | | | | |
| Llama-3-70B-Instruct | 8.35 | 49.20 | 2.00 | 19.85 |
| AgentLM-13B | 39.50 | 2.75 | 0.00 | 14.08 |
| AgentLM-70B | 49.50 | 10.68 | 4.00 | 21.39 |
| Agent-FLAN | 40.35 | 28.64 | 16.00 | 28.33 |
| Llama-3-8B-Instruct | 13.49 | 14.48 | 10.00 | 12.66 |
| w/ Reflexion | 14.08 | 12.55 | 8.00 | 11.54 |
| w/ Iterative SFT | **78.21** | 41.42 | 55.00 | 58.21 |
| w/ **CGI** (*Ours*) | 76.17 | **78.43** | **68.00** | **74.20** |

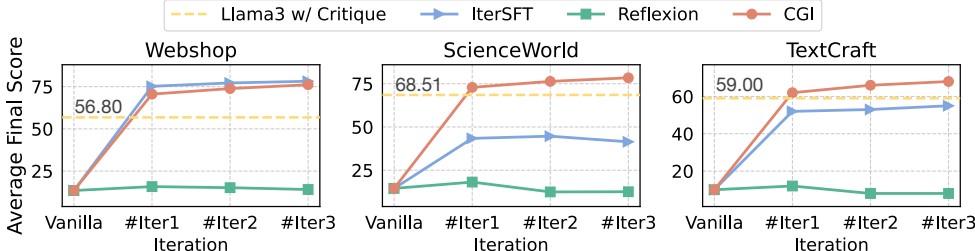

Figure 2: Performance of different iterative methods across iterations in three agentic tasks.

Model shows minimal sensitivity to the base model, achieving 68.51% with Llama-3-8B and 72.44% with Llama-3-70B. However, the self-critique approach, which also relies on verbal feedback, shows a negative effect on both Llama-3-70B, 8B and its fine-tuned variant. This suggests that while verbal feedback can convey richer information, untrained self-critique methods often produce low-quality or unstructured feedback, failing to provide effective guidance.

**Finding 2: Fine-tuned models struggle to utilize critiques.** While supervised fine-tuning significantly improves baseline performance (+33.11%), our experiments reveal a key limitation: fine-tuned models struggle to effectively incorporate critique feedback. In the ScienceWorld scenario, Vanilla Llama3-8B and Llama3-70B show substantial improvements with Critic guidance, achieving average scores of 68.51% and 72.44%, respectively. However, despite having a stronger base performance, fine-tuned Llama3-8B only reaches 55.94% with the same critique mechanism. This issue persists in the WebShop scenario, where critique guidance even causes a performance decline in fine-tuned Llama3-8B, dropping from 76.12% to 74.68%. These results suggest that while fine-tuning enhances base performance, it may reduce the model's responsiveness to external feedback. For example, in the ScienceWorld scenario, despite the critic model advising the actor to "look around" and assess the environment for possible actions, the actor persists in executing "Go to Bathroom", an action that seems closest to the task description. This behavior disregards the fact that the actor cannot reach the bathroom from its current position, leading to a deadlock (see Appendix H.1). Our iterative action refinement approach helps mitigate this issue; after three iterations, the model shows a significant improvement in critique utilization, with its performance surpassing that of the actor model with only supervised fine-tuning by +15.99%, achieving state-of-the-art results compared to various baselines.

**Finding 3: CGI continuously enhance model performance via action refinement.** As shown in Figure 2, compared to other iterative methods (*e.g.*, Reflexion and vanilla iterative SFT), CGI consistently supports model performance improvement. In contrast, Reflexion shows minimal improvement and can even cause performance degradation. This aligns with previous findings [20], where self-critique struggles to escape local minima in agentic tasks that require significant diversity and exploration. Iterative SFT achieves a notable improvement only in the first iteration, with minimal progress in the following 2-3 iterations. This issue is more pronounced in long-horizon

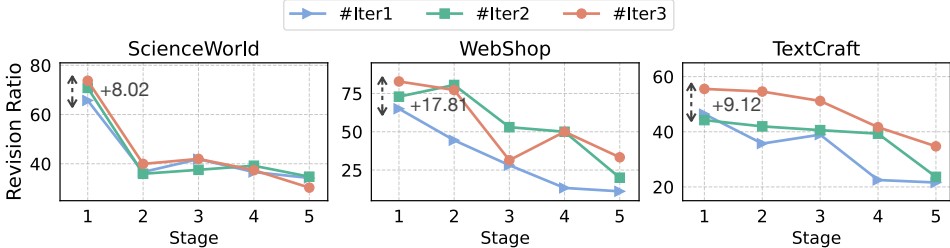

Figure 3: Revision Ration of actor model at different trajectory stages across three tasks. #Iter-$k$ denotes the $k$-th iteration of action refinement. Revision Ratio represents the proportion of actions that differ after critique compared to those without critique.

tasks (*e.g.*, ScienceWorld and TextCraft), suggesting that the model may over-sample simpler tasks while under-sampling more complex ones. Consequently, the training set for subsequent iterations becomes dominated by easier tasks, with few solutions for harder ones. As iterations progress, this bias deepens, leading to a long-tail distribution where solutions to more challenging tasks are scarce, ultimately causing the model to plateau or even degrade. In contrast, CGI leverages high-quality critiques and enhances the actor's utilization, enabling the model to consistently sample solutions to more difficult tasks, thus supporting sustained performance improvement.

## 7 Further Analysis

### 7.1 Qualitative Analysis of CGI

In this section, we conduct a qualitative analysis to examine how CGI enhances performance in agentic tasks. Specifically, we focus on two key aspects: *1)* **Trajectory-level**: Identifying the stages in the trajectory where CGI provides the most significant improvement; *2)* **Task-level**: Evaluating CGI's performance across tasks of varying trajectory length (see Appendix E for categorization).

**CGI significantly improves early-stage performance.** To determine at which stage of the trajectory CGI contributes most to performance improvements, we divide the trajectory into five stages based on its length and compute the extent to which the critique influences the actor model's behavior, measured by the Revision Ratio. As shown in Figure 3, across three tasks, actor models exhibit the highest action revision frequency in stage 1, with the revision ratio dropping sharply in later stages. This suggests that the critique primarily guides the actor during early exploration, helping reduce ineffective searches. Notably, as the actor undergoes more refinement iterations, its revision ratio in stage 1 increases (+8.02% on ScienceWorld, +17.81% on WebShop, +9.12% on TextCraft), indicating that better critique utilization accelerates effective exploration. This, in turn, enhances CGI's efficiency, enabling it to achieve higher scores in fewer steps (Figure 8).

**CGI helps the model to continuously improve on long-horizon tasks.** To investigate the effect of CGI on tasks of varying trajectory length, we categorize the ScienceWorld scenarios into three difficult-level groups based on the average length of the oracle agent's trajectories. Longer trajectories correspond to higher difficulty levels. As shown in Figure 4, the vanilla model (Llama3-8B) performs poorly, with performance decreasing as the task length increases. Critique-Guided Inference leads to a significant improvement, particularly on easy tasks, where performance increases by +59.64%. Although there is also a notable improvement on harder tasks (+43.95%), the gain is less pronounced compared to easier tasks. However, with iterative action refinement, the model exhibits the greatest improvement on hard tasks, with a +28.75% increase after three iterations. These results suggest that models

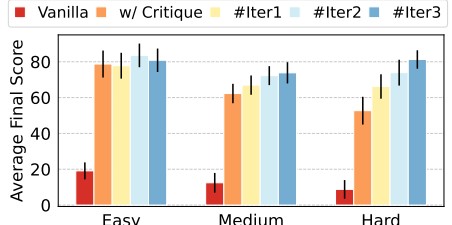

Figure 4: Performance of models across varying trajectory length. Vanilla refers to the Llama3-8B model, w/ Critique refers to the critique-guided inference with our trained Critic-Llama3, and #Iter-$k$ represents the $k$-th iteration of action refinement.

without action refinement can quickly adapt to critiques for easy and medium-level tasks. However, for longer tasks, the model's ability to leverage critiques diminishes. Action refinement helps the model continuously follow critiques, thereby enhancing performance on long-horizon tasks.

## 7.2 Effects of Number of Candidate Actions

To further evaluate the scalability of CGI, we varied the number of candidate actions ($M$) sampled at each step during inference. As shown in Figure 5, we observe the following trends: CGI achieves a significant improvement with just one candidate action, reaching an average score of 56.89%. As $M$ increases, performance improves further, rising from 56.89% to 61.72%. However, when $M = 7$, performance plateaus, indicating saturation. Other numerical- and verbal-based methods exhibit similar trends. For GPT-4 as a critic, performance increases with $M$, improving from 32.28% to 38.13%. DGAP is particularly sensitive to the number of candidate actions, with performance rising from 15.02% to 25.07%. This highlights that numerical methods are highly dependent on the quality of the sampled candidate actions, as they only discriminate but not suggest meaningful revisions. In contrast, self-critique does not benefit from an increase in $M$. This suggests that low-quality critiques can degrade performance, regardless of the number of candidate actions.

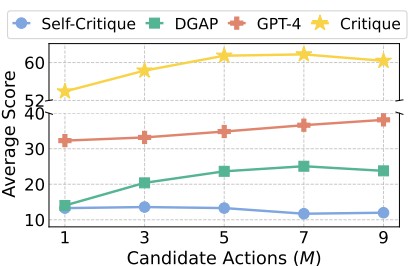

Figure 5: Performance of different methods across varying candidate actions. The average score is the mean final score across three environments.

## 8 Ablation Study

To analyze the contribution of the three types of data (*i.e.*, $\mathcal{D}_{correct}$, $\mathcal{D}_{refine}$, and $\mathcal{D}_{general}$) used to enhance the actor model during the action refinement stage, we conduct an ablation study. Specifically, we remove each dataset from the training data used in the first iteration of action refinement and evaluate the final performance under the guidance of the critic model. As shown in Figure 6, removing any one of the three datasets results in a performance drop compared to the actor model trained with the full dataset (CGI $_{\#Iter1}$). The most significant decrease occurs when the critique-action pairs are removed (w/o $\mathcal{D}_{refine}$), where the average performance across the three tasks drops from 68.50% to 50.37%. This decline is more pronounced for longer tasks, with performance reductions of 5.4% on WebShop, 22% on TextCraft, and 26.94% on ScienceWorld. These results highlight that enhancing the actor's critique utilization ability is the most critical factor in improving overall performance. A less severe drop is observed when the generalization data set is excluded (w/o $\mathcal{D}_{general}$), highlighting the importance of general instructions for model generalization, which helps the model adapt to the unseen test set.

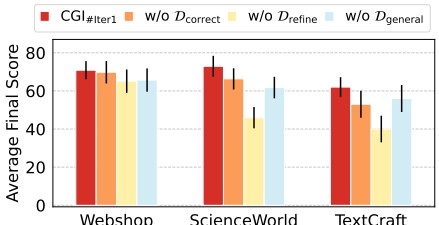

Figure 6: Ablation study of the three data types used in action refinement: $\mathcal{D}_{correct}$ (correct trajectories to improve the actor's reasoning ability), $\mathcal{D}_{refine}$ (critique-action pairs to enhance critique utilization), and $\mathcal{D}_{general}$ (data to maintain the model's generalization capability). CGI $_{\#Iter1}$ refers to the actor model trained in the first iteration of action refinement using the full dataset.

## 9 Conclusion

We have introduced **C**ritique-**G**uided **I**mprovement (CGI), a two-player framework that emphasizes nature language feedback for iterative refinement of LLM-based agents. By separating the roles of an actor, which proposes actions, and a critic, which provides verbal guidance, CGI circumvents the limitations of purely numerical signals and addresses the challenges arising from self-refinement. Experimental results in three interactive and agentic environments confirm the effectiveness of CGI, with a small critic model outperforming GPT-4 in providing feedback.

## Acknowledgement

We appreciate the support from the Chinese NSF General Program (No.62572129), Major Research Plan (No.92270121). We also acknowledge the use of an icon from Flaticon[5] and thank its creators for providing this visually appealing design.

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

# Appendix

## A   Limitations

CGI incurs higher computational time compared to baseline methods but remains within a reasonable range. As shown in Table 3, CGI requires approximately four times the computational time of the baseline. However, this additional cost results in a substantial performance improvement, with CGI achieving a 43.31% gain over No Critique. This indicates that the extra computational time is effectively utilized. In contrast, other verbal-based methods, such as GPT-4o and Self-Critique, do not produce significant improvements and require more computational time than CGI.

## B   Broader Impacts

**Positive Societal Impacts.**   By generating and leveraging natural language critiques, CGI enhances the performance in embodied AI tasks (*e.g.*, , ScienceWorld) and real-world scenarios (*e.g.*, , Web-Shop), leading to improved task performance. This contributes to enabling agents to better understand and align with human goals, ultimately supporting more helpful, interpretable AI systems. Moreover, our critic model demonstrates strong performance despite using a relatively small backbone (Llama-3-8B), which significantly reduces the computational and environmental costs compared to larger proprietary models. This also promotes broader accessibility to advanced feedback mechanisms.

**Negative Societal Impacts and Risks.**   Natural-language critiques may inadvertently incorporate biases from the training data or expose private information through the agent's reasoning process. Moreover, as these critiques provide explicit, step-by-step explanations, they could be misused by bad actors to facilitate disinformation, social engineering, or the generation of unsafe code. The relatively small size of the critic model further reduces the cost of replication and misuse, increasing the associated risks.

## C   Notations

| Symbol | Dataset Definitions | |
|---|---|---|
| | **Meaning** | |
| $\mathcal{D}_{\text{critique}}$ | Expert critique dataset generated by expert critic (GPT-4o), used for training the critic model as described in §4.2 | |
| $\mathcal{D}_{\text{correct}}$ | Correct trajectory dataset collected during each iteration of the action refinement process | |
| $\mathcal{D}_{\text{refine}}$ | Critique-action pairs dataset used for learning how to utilize critiques effectively | |
| $\mathcal{D}_{\text{general}}$ | General conversational dataset (*e.g.*, ShareGPT) used to maintain general language modeling capabilities during fine-tuning | |

| Symbol | Loss Function Parameters | |
|---|---|---|
| | **Meaning** | |
| $\mathcal{L}_{\text{actor}}$ | Actor model loss function with three components: learning from expert/correct trajectories, learning to utilize critiques, and maintaining general capabilities | |
| $\beta$ | Weight parameter balancing agentic task learning vs. general capabilities (typically $\beta = 0.8$) | |
| $y$ | General text responses from datasets like ShareGPT for maintaining language modeling abilities | |

# D Instruction Prompt Examples

The system prompts for three agentic environments are presented in Listing 1. The instruction for critique generation introduced in §4.2 is presented in Listing 2.

Listing 1: Prompt details for ScienceWorld, WebShop, and TextCraft.

```
ScienceWorld Instruction:
You are an agent for the science world. Every round I will give you an
observation, you have to respond with an action based on the observation
to finish the given task.
Here are the actions you may take:
{"action": "open/close OBJ", "description": "open/close a container",}
{"action": "de/activate OBJ", "description": "activate/deactivate a
device",}
{"action": "connect OBJ to OBJ", "description": "connect electrical
components", }
{"action": "disconnect OBJ", "description": "disconnect electrical
components",}
{"action": "use OBJ [on OBJ]", "description": "use a device/item",}
{"action": "look around", "description": "describe the current room",}
{"action": "look at OBJ", "description": "describe an object in detail",}
{"action": "look in OBJ", "description": "describe a container's contents
",}
{"action": "read OBJ", "description": "read a note or book",}
{"action": "move OBJ to OBJ", "description": "move an object to a
container", }
{"action": "pick up OBJ", "description": "move an object to the inventory
", }
{"action": "put down OBJ", "description": "drop an inventory item",}
{"action": "pour OBJ into OBJ", "description": "pour a liquid into a
container", }
{"action": "dunk OBJ into OBJ", "description": "dunk a container into a
liquid", }
{"action": "mix OBJ", "description": "chemically mix a container",}
{"action": "go to LOC", "description": "move to a new location",}
{"action": "eat OBJ", "description": "eat a food",}
{"action": "flush OBJ", "description": "flush a toilet",}
{"action": "focus on OBJ", "description": "signal intent on a task object
",}
{"action": "wait", "description": "take no action for 10 iterations",}
{"action": "wait1", "description": "take no action for 1 iteration", }
{"action": "task", "description": "describe current task",}
{"action": "inventory", "description": "list your inventory"}
Your response should use the following format:

Thought: your thoughts.
Action: your next action

WebShop Instruction:
You are web shopping. I will give you instructions about what to do. You
have to follow the instructions. Every round I will give you an
observation and a list of available actions, you have to respond an
action based on the state and instruction.
You can use search action if search is available. You can click one of
the buttons in clickables. An action should be of the following structure
: search[keywords] click[value]. If the action is not valid, perform
nothing. Keywords in search are up to you, but the value in click must be
 a value in the list of available actions. Remember that your keywords in
 search should be carefully designed. Your response should use the
following format:

Thought: I think ...
Action: click[something] or search[something]
```

```
TextCraft Instruction:
You are given a few useful crafting recipes to craft items in Minecraft.
Crafting commands are of the format "craft [target object] using [input
ingredients]". Every round I will give you an observation, you have to
respond to an action based on the state and instruction. You can "get" an
 object (ingredients) from the inventory or the environment, look up the
game "inventory" by inventory, or "craft" (target) using any of the
crafting commands. You can use ONLY these crafting commands provided, do
not use your own crafting commands. However, if the crafting command uses
 a generic ingredient like "planks", you can use special types of the
same ingredient e.g. dark oak "planks" in the command instead. Your
response should use the following format:

Thought: ...
Action: ...
```

Listing 2: The instruction prompts for Critique Generation.

```
Critique Generation Instruction:
Your task is to critique the candidate's next-step action based on the
agent's task goal and interaction history.

{available_actions}

Critique Steps
Step 1: Analyze Candidate Action
Examine the candidate's action based on the following criteria and assign
 an overall grade using this scale: Excellent, Good, Neutral, Poor, Very
Poor.

Critique Dimensions
- Contribution: Assess whether the action contributes to completing the
agent's task. This includes both direct actions (e.g., picking up the
target OBJ) and indirect actions (e.g., reasonable exploration that can
provide additional environmental information and facilitate future
progress).
- Feasibility: Assess whether the action is valid according to the agent'
s predefined Allowed Action Types list.
- Efficiency: Analyze whether the action optimally achieves the task
without unnecessary steps or redundancy.

Step 2: Provide Revision Suggestions
Suggest a modification to align the candidate's action better with the
task or the agent's action capabilities. Note that the suggested revision
 should be based on the Allowed Action and Object Types.

Critique Format
Please structure your critique in the following format:
## Contribution: [Analysis of Contribution].
## Feasibility: [Analysis of feasibility].
## Efficiency: [Analysis of efficiency].
## Overall Grading: [Overall grade: Excellent/Good/Neutral/Poor/Very Poor
].
## Suggested Revision: [Brief revision suggestion, if applicable].

Inputs:
The agent's task goal and interaction history:

{history}

Candidate next step action: {candidate_action}

Now, please provide your critique:
```

**Critique Generation Instruction** (with expert path):
Your task is to critique the candidate next-step action based on the agent's task goal and interaction history. The gold path for current task is provided as a reference to guide your critique, but do not explicitly mention it in your critique.

{available_actions}

**Critique Steps**

**Step** 1: Analyze Candidate Action
Examine the candidate's action based on the following criteria, then assign an overall grade using this scale: Excellent, Good, Neutral, Poor, Very Poor.

**Critique Dimensions**
-Contribution: Assess whether the action contributes to completing the agent's task. This includes both direct actions (e.g., picking up the target OBJ) and indirect actions (e.g., reasonable exploration that can provide additional environmental information and facilitate future progress).
-Feasibility: Assess whether the action is valid according to the agent's predefined Allowed Action Types list.
-Efficiency: Analyze whether the action optimally achieves the task without unnecessary steps or redundancy.

**Step** 2: Provide Revision Suggestions
Suggest a modification to align the candidate's action better with the task or the agent's action capabilities. For example, if the action is not allowed, recommend an alternative from the action list that aligns better with the task goal.

**Critique Format**
Please structure your critique in the following format:
## Contribution: [Analysis of Contribution].
## Feasibility: [Analysis of feasibility].
## Efficiency: [Analysis of efficiency].
## Overall Grading: [Overall grade: Excellent/Good/Neutral/Poor/Very Poor].
## Suggested Revision: [Brief revision suggestion, if applicable].

Referenced Gold Path for Current Task:

{gold_path}

**Inputs**:
The agent's task goal and interaction history:

{history}

Candidate next step action: {candidate_action}

Now, please provide your critique:

Table 3: Computational efficiency of different methods in three agentic environments per task.

| Method | WebShop | | Sciworld | | Textcraft | |
|---|---|---|---|---|---|---|
| | Inference Time | Performance | Inference Time | Performance | Inference Time | Performance |
| No Critique | 0.38 min | 13.49 | 1.49 min | 14.48 | 1.02 min | 10.00 |
| w/ DGAP | 1.13 min | 30.41 | 3.13 min | 19.52 | 1.33 min | 21.00 |
| w/ Self-Critique | 1.36 min | 1.50 | 3.42min | 10.06 | 2.17 min | 19.00 |
| w/ GPT-4o | 2.34 min | 17.78 | 6.28 min | 33.06 | 3.28 min | 46.00 |
| w/ Critic Model | 1.22 min | **56.80** | 3.41 min | **68.51** | 1.98 min | **59.00** |

# E   Evaluation Settings

In the experiments presented in this paper, we used three agentic environments: WebShop, SciWorld, and TextCraft.

## E.1   Details of Environments

**WebShop**   WebShop is an interactive web environment designed for web shopping. In this environment, agents are given instructions and must purchase a product that meets specified criteria. Agents can either click a button on the webpage or use the search engine to find the product. We evaluate performance using the success rate, with a maximum of 10 rounds per task.

**ScienceWorld**   ScienceWorld is a benchmark environment for testing agents' scientific reasoning abilities, based on a standard elementary science curriculum. It includes 30 types of tasks, such as using measurement instruments and conducting mechanics experiments. The action space is task-specific, with the environment simulator providing the effects of actions. We use reward as the evaluation metric, with a maximum of 30 rounds per task. Task lengths are classified based on the ScienceWorld environment, which provides gold-standard trajectories from 30 hand-coded oracle agents. We adopt the following categorization: Short (11.76), Medium (28.58), and Long (94.30). These lengths correspond to the average number of steps taken by the oracle agent.

**TextCraft**   TextCraft is a text-only environment for crafting Minecraft items. It constructs a crafting tree based on Minecraft's crafting recipes, consisting of 544 nodes, each representing a target item. For each task, the agent is given a target item and a list of crafting commands generated by the tree. Tasks vary in complexity, ranging from 1 to 4 steps. The environment supports three valid actions:"craft <item> using <ingredients>", "get <item>", and "inventory". After each round, the environment checks the agent's actions and returns the execution state. Agents are rewarded with 1 point only upon successfully crafting the target item. We evaluate performance using the success rate, with a maximum of 20 rounds per task.

## E.2   Computational Efficiency Analysis

We evaluated the computational efficiency of CGI by conducting experiments in three agentic environments, using Llama-3-8B as the actor model. We measured the average inference time per task. As shown in Table 3, although CGI incurred higher computation times compared to inference without critique guidance (i.e., No Critique), these times remained within an acceptable range. Furthermore, CGI demonstrated a substantial reduction in inference time compared to using GPT-4 as the critic, while achieving significant performance improvements.

# F  Implementation Details

In our experiments, we use the Llama-3-8B-Instruct [3] as the base model for both the actor and the critic.

## F.1  Training Data

**Critique Generation**   The training data for the critic model consists of expert critiques generated by the expert critic (*i.e.*, GPT-4o) in the ScienceWorld, Webshop, and TextCraft environments, as described in § 4.2. The specific training set sizes during the SFT phases for Llama3-8B-Instruct are 14K from ScienceWorld, 10K from Webshop, and 8K from TextCraft. Examples of our SFT training data are provided below:

```
SFT Example:
{"conversations":
[{"from": "human", "value": "Your task is to critique the candidate's
next-step action based on the agent's task goal and interaction history.
Allowed Action Types:- search[KEYWORDS]: perform a web search - click[OBJ
]: click an object or link on a webpage. Step 1: Analyze Candidate Action
.Examine the candidate's action based on the following criteria and
assign an overall grade using this scale: Excellent, Good, Neutral, Poor,
 Very Poor. Critique Dimensions: Contribution: Assess whether the action
contributes to completing the agent's task. This includes both direct
actions and indirect actions. Feasibility: Assess whether the action is
valid according to the agent's predefined action list. Specifically,
check if the action is part of the Allowed Action Types, whether the OBJ/
KEYWORDS within the action is clearly specified, and whether it is
formatted correctly according to the system's requirements. Efficiency:
Analyze whether the action optimally achieves the task without
unnecessary steps or redundancy. Step 2: Provide Revision Suggestionn:
Suggest a modification to align the candidate's action better with the
task or the agent's action capabilities. For example, if the action is
not allowed, recommend an alternative from the action list that aligns
better with the task goal. Critique Format:Please structure your critique
 in the following format : ## Contribution: [Analysis of Contribution].
## Feasibility: [Analysis of feasibility]. ## Efficiency: [Analysis of
efficiency].## Overall Grading: [Overall grade: Excellent/Good/Neutral/
Poor/Very Poor]. ## Suggested Revision: [Brief revision suggestion, if
applicable]. ### InputsThe Agent's Task Goal and Interaction History:
WebShop [SEP] Instruction: [SEP] Find me machine wash, moisture wicking
men's t-shirts & tanks with polyester spandex [SEP] Search Candidate Next
 Step Action: search[machine wash moisture wicking men's t-shirts with
polyester spandex long sleeve red blue XXL price: under 50]. Now, please
provide your critique:"},
{"from": "gpt", "value": "## Contribution: The candidate's action
contributes to the task by attempting to perform a search for the
specified product. It includes most of the necessary attributes such as
material, sleeve length, color, size, and price, which are crucial for
finding the desired item. ## Feasibility: The action is feasible as it
uses the allowed action type "search[KEYWORDS]". However, there are some
discrepancies in the keywords used. The candidate action uses "XXL"
instead of "xx-large" and "price: under 50" instead of "price < 50.00".
While these variations might still yield relevant results, they do not
exactly match the specified format. ## Efficiency: The action is
relatively efficient as it attempts to gather relevant search results in
one step. However, the slight variations in keyword formatting could
potentially lead to less precise search results, which might require
additional steps to refine the search. ## Overall Grading: Good. ##
Suggested Revision: Modify the search keywords to exactly match the
specified format: search[machine wash moisture wicking men's t-shirts &
tanks polyester spandex long sleeve red blue size xx-large price <
50.00]. This ensures consistency and potentially more accurate search
results."}],
"system": ""}}
```

Table 4: Training dataset size for each iteration of Action Refinement stage.

| Env | Gold | | #Iter 1 | | #Iter 2 | | #Iter 3 | |
|---|---|---|---|---|---|---|---|---|
| | Path | Critique | Path | Critique | Path | Critique | Path | Critique |
| **ScienceWorld** | 378 | 4106 | 133 | 1464 | 158 | 1803 | 193 | 2205 |
| **WebShop** | 427 | 1659 | 106 | 663 | 283 | 1637 | 344 | 2010 |
| **TextCract** | 441 | 1701 | 233 | 1423 | 310 | 2070 | 340 | 2049 |
| **OverAll** | 1246 | 6740 | 472 | 3550 | 751 | 5510 | 877 | 6264 |

**Action Refinement**   As described in § 4.3, we performed action refinement three times. Each training dataset consists of two parts: Correct Path, which includes the correct trajectories obtained by the actor from interactions with the environment in the previous iteration; and Correct Critique, which includes the <critique, action> pairs that the actor correctly followed during the previous iteration. The dataset size for each iteration of Llama-3-8B-Instruct is detailed in Table 4.

## F.2   Finetuning Details

We ran SFT experiments using 8 NVIDIA A100-40GB GPUs. For action refinement, to prevent policy drift, we only trained the base model (*i.e.*, , Llama-3-8B-Instruct) each time. We conduct experiments with the LlamaFactory code base[6]. The configurations of our hyper-parameters are detailed in Table 5.

Table 5: Fine-tuning hyper-parameters for Critique Generation and Action Reinement stage.

| Configuration | Critique Generation | Action Refinement |
|---|---|---|
| Model | Llama-3-8B-Instruct | Llama-3-8B-Instruct |
| Number of epochs | 3 | 3 |
| Devices | 8 A100 GPU (40 GB) | 8 A100 GPU (40 GB) |
| Total Batch size | 64 samples | 64 samples |
| Optimizer | Adam [58] | Adam [58] |
| | $(\beta_1 = 0.9, \beta_2 = 0.98, \epsilon = 1 \times 10^{-8})$ | $(\beta_1 = 0.9, \beta_2 = 0.98, \epsilon = 1 \times 10^{-8})$ |
| Learning rate | $2 \times 10^{-5}$ | $2 \times 10^{-5}$ |
| Warmup Ratio | 0.05 | 0.05 |
| Cutoff Length | 4096 | 4096 |
| Training Time | 5h 46m 24s | 11h 11m 55s |

## F.3   Effect of Iteration Count

Here we analyze the impact of the number of iterations on the performance of CGI. As shown in Figure 7, the fourth iteration leads to a performance drop in ScienceWorld and TextCraft, while the improvement in WebShop is marginal (+0.67%). The third iteration achieves the highest average performance across all tasks. These results suggest that additional iterations do not yield further gains. Therefore, we report the results from the third iteration in the main results section (§ 6).

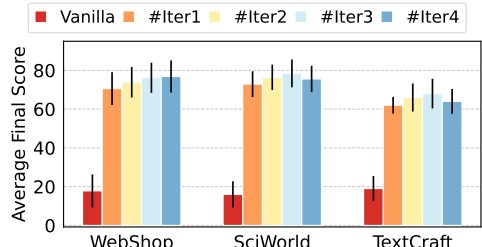

Figure 7: Performance of models with varying iteration counts across three tasks.

## F.4   Effect of Training with Expert Trajectories and Updating the Critic

In this section, we analyze two key factors: *1)* whether our critic heavily depends on expert trajectories; *2)* whether updating the critic model through iterations can continue to improve performance. To evaluate the first factor, we replace the gold-standard expert trajectories with GPT-4o-generated

---

[6]https://github.com/hiyouga/LLaMA-Factory

Table 6: Performance comparison across different methods and datasets

| Method | WebShop | ScienceWorld | TextCraft | Average |
|--------|---------|--------------|-----------|---------|
| Llama3-8B | 13.49 | 14.48 | 10.00 | 12.66 |
| w/ GPT-4o | 17.78 | 33.06 | 46.00 | 32.28 |
| w/ Critic (Trained on GPT-4o Traj) | 43.21 | 56.93 | 52.00 | 50.71 |
| w/ Critic (Trained on Expert Traj) | 56.80 | 68.51 | 59.00 | 61.44 |
| w/ Updated Critic | 62.51 | 74.56 | 64.00 | 67.02 |

trajectories across three environments, following the exact pipeline described in §4.2 for training the critic model. For the second factor, we perform iterative updates on the critic model after action refinement to adapt to the updated actor's policy. Specifically, we use the critic from the first iteration to generate new critique data and retrain the critic.

As shown in Table 6, CGI remains highly effective even without gold-standard expert data. Our method achieves an average performance of 50.71, significantly outperforming both the baseline and the GPT-4o model that generated the initial data. This highlights the robustness of our framework. Additionally, the critic can be dynamically updated. The results indicate that updating the critic leads to further performance improvements (from 61.44% to 67.02%), demonstrating that the critic is not limited to being static and can evolve alongside the actor.

## G Baselines

### G.1 Critique Methods

In Section 6, we compare our trained Critic-Llama3 model with other verbal- and numerical-based methods. For the verbal-based approaches, such as self-critique and GPT-4, we generate critiques by prompting the model with the instructions provided in Appendix D. For the numerical-based method (i.e., DGAP), following [14], we compute the cosine similarity between the actor model (Llama-3-8B) and expert data at each step. We collected 23K data points for Scienceworld, 16K for WebShop, and 12K for TextCraft. Examples of the reward model training data are provided below:

```
DGAP Example:
{"input": "Your task is to measure the melting point of lead, which is
located around the kitchen. 10. look around. Action: go to hallway", "
Score": "10"}
{"input": "Your task is to measure the melting point of lead, which is
located around the kitchen. 10. look around. Action: look at art studio",
 "Score": "0"}
{"input": "Your task is to measure the melting point of lead, which is
located around the kitchen. 10. look around. 9. go to hallway. Action:
put down orange", "Score": "0"}
{"input": "Your task is to measure the melting point of lead, which is
located around the kitchen. 10. look around. Action: look at hallway", "
Score": "9.03"}
{"input": "Your task is to measure the melting point of lead, which is
located around the kitchen. 10. look around. 9. look at hallway. Action:
open door to outside", "Score": "6.13"}
{"input": "Your task is to measure the melting point of lead, which is
located around the kitchen. 10. look around. 9. look at hallway. 8. open
door to outside. Action: teleport to kitchen", "Score": "8.87"}
```

### G.2 Iterative Methods

In Section 6, we compare CGI with two iterative baselines: Reflexion and Iterative SFT. Following [20], we prompt the model for self-reflection (Listing 3) at the end of each iteration. The insights gained from the current round are then incorporated into the system prompt to guide the model's decision-making in the subsequent round. For Iterative SFT, we use the actor model $\pi_\theta^k$ from the previous round to interact with the environment. The newly collected correct trajectories are combined with the expert path to form the training data for the next round. Specifically, for iterations 1 to 3, we

used 1,676, 1,719, and 1,750 trajectories, respectively, mixing them with the general dataset (*i.e.*, ShareGPT) at a 1:4 ratio for training. To prevent policy drift, we update $\pi_\theta^0$ at each iteration. Each iteration consists of 3 epochs. The learning rate for Iterative SFT is set to 2e-5, with a 3% warm-up and a cosine scheduler. We use the AdamW optimizer with a weight decay of 0 and apply a maximum gradient clipping norm of 1 to prevent gradient explosion. The batch size is set to 2 per device, with a sequence length of 4,096 and an accumulation count of 4, adjusted based on sequence parallelism.

Listing 3: The instruction prompt for Reflexion.

```
Reflexion Instruction:
Your task is to provide a next plan of action based on the previous
attempt's interaction history. The next plan should reflect the lessons
learned from the failure and adjust for better results.

Previous Trial:
{history}

Next Plan:
```

# H  Case Study

## H.1  Failure Cases of Fine-tuned Llama3

We provide negative examples where the fine-tuned Llama3-8B does not follow the critique.

Listing 4: Failure Cases of Fine-Tuned Llama3-8B in ScienceWorld

```
Task Description:
Your task is to measure the temperature of unknown substance B, which is
located around the bathroom. First, focus on the thermometer. Next, focus
 on the unknown substance B. If the unknown substance B temperature is
above 100.0 degrees celsius, place it in the orange box. If the unknown
substance B temperature is below 100.0 degrees celsius, place it in the
yellow box. The boxes are located around the bathroom.
History:
Action: open door to kitchen
Observation: The door is now open.
Action: go to kitchen
Observation: You move to the kitchen.
Action: open door to hallway
Observation: The door is now open
Action: go to hallway
Observation: You move to the hallway.
Critique:
......
### Feasibility: The candidate action "open door to bathroom" is
efficient in terms of task progression, as it directly addresses the need
 to access the bathroom. However, the agent should consider checking the
current state first to ensure the action is feasible by "Look Around".
### Overall Grading: Netual
......
Action: go to bathroom
Observation: No known action matches that input.
```

## H.2  Critic-Llama3 Outperforms GPT-4o

We provide the following examples to explain why our critic outperforms GPT-4o when the actor model is fixed as Llama3-8B. Compared to GPT-4o, our critic demonstrates superior spatial reasoning and a higher degree of goal-directed behavior.

Listing 5: Comparison between Critic-Llama3 and GPT-4o as Critics

```
Example 1: Spatial Reasoning
```

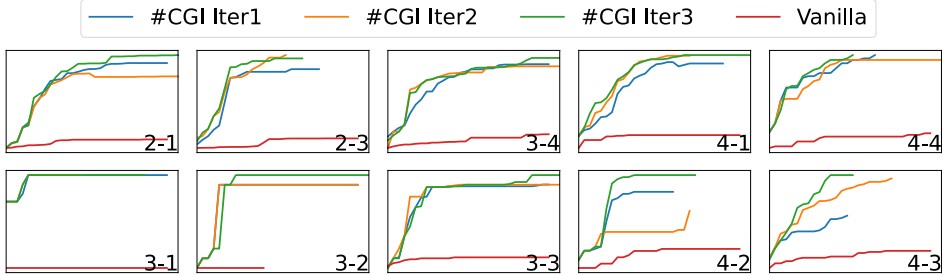

Figure 8: Visualizing trajectories of CGI, and base model (Llama-3-8B-Instructt), the x-axis represents time steps, ranging from 0 to T, while the y-axis denotes scores, which vary from 0 to 100. Each graph illustrates the trajectories corresponding to different tasks in test variation. Task identifiers are positioned at the bottom right of each figure.

```
State: Agent is in the workshop. Task requires an object in the bathroom.
Action: focus on thermometer
GPT-4o (Incorrect): Overall Grading: Excellent.
It approves the action based on the task description, ignoring the agent'
s location.
Our Critic (Correct): Overall Grading: Very Poor.
It correctly identifies the thermometer is not in the current room and
suggests navigating first.

Example 2: Goal-Directedness
Task: Find a living thing.
Action: move to the closet (an exploratory but inefficient action).
GPT-4o (Inefficient): Overall Grading: Neutral.
It allows this exploratory step.
Our Critic (Efficient): Overall Grading: Poor.
It rejects the inefficient action and suggests a more direct one, like
opening the door to explore other rooms.
```

## H.3  Trajectory Visualization of ScienceWorld

We visualize the cumulative scores of the actor performing different numbers of action refinements (#Iter $k$) and the base model in the ScienceWorld scene[7]. As shown in Figure 8, our CGI method shows a notable improvement in efficiency, achieving higher scores in fewer steps. Furthermore, as the number of action refinements increases, efficiency also improves.

---

[7]Detailed information of each task can be found in https://github.com/allenai/ScienceWorld

