# OpenReview forum: "The Lighthouse of Language: Enhancing LLM Agents via Critique-Guided Improvement"
_NeurIPS.cc/2025/Conference — NeurIPS 2025 poster_

### Official Review · Reviewer_v7iq · 2025-06-04

**Clarity:** 3
**Significance:** 3
**Originality:** 2
**Rating:** 4
**Confidence:** 4

**Summary:**

The authors propose to improve an LLM agent using a two-player approach. First, they pre-train an LLM critic to evaluate the actions of an LLM actor and suggest refinements to its actions. Then, they fine-tune the LLM actor so that it proposes better actions next time. They compare their approach to 7 LLM training baselines and 9 zero-shot models in three textual environments and provide a few side analyses and ablations.

**Questions:**

- The proposed framework is very close to SELU [1], apart from the fact that SELU addresses multimodal LLMs rather than purely text-based ones. In SELU they also have a critic-guided improvement mechanism.
[1]:Li, B., Jiang, H., Ding, Z., Xu, X., Li, H., Zhao, D., & Lu, Z. (2024). SELU: Self-Learning Embodied MLLMs in Unknown Environments. arXiv preprint arXiv:2410.03303.

- The impact of initializing the training dataset with expert trajectories is not studied. An ablation where this initialization is removed is missing, as the expert paths may play a key role in the good performance of the method, but the need for collecting such expert paths is a key weakness of the approach. Can the authors provide such an ablation in their rebuttal?

- To insist on this potential weakness, if the authors have a way to generate expert trajectories, why learn another agent that will perform the same trajectories? Is it just a matter of distilling into a smaller size agent?

- The results presented in Tables 1 and 2 seem to result from a single run for each condition. Is there some variability in the results if one performs the same experiment several times? Is the experiments are too costly to be run several times, could the authors measure the variability on a small subset of the setup?

- The comparison between verbal feedback and numerical feedback is potentially very interesting but the proposed study does not adequately focus on this specific question, as it might also be the critic accuracy that is different between DGAP, Explicit RM and CGI, and not only the feedback modality.

- Expanding on the above point, isn't the point that CGI proposes (potentially not proposed by the actor) action refinements and the actor is fine-tuned on the difference to these actions, whereas in DGAP and Explicit RM, the actor will be fine-tuned on the difference to the best-ranked action among the ones proposed by the actor?

*More local points*

- line 117: why not also train the agents from refined versions of failed trajectories? How is performance influenced by this design choice of only training on successful trajectories?

- line 145: To avoid overfitting, we follow previous work [29] by fine-tuning the original model πθ rather than the previous iteration model... -> Does this mean that the datasets are incremented at each iteration with the content of the previous iterations, or does CGI purely forget all improvements from the previous iterations?

- did the authors make sure that trajectories from the test set are never seen in the training set?

- "our 8B critic model significantly outperforms GPT-4o on both the Llama-3 (8B and 70B) and Llama-3-8B’s fine-tuned variant trained on expert data." -> the sentence does not make sense to me.

- would it make sense to merge Tables 1 and 2 into a single table?

- line 275 : are the authors sure that longer trajectories are always the most difficult? It is not the case in general...

- putting all related work in the Appendix is a bad practice

*Typos:*
- In Fig. 1: Dimentions -> Dimensions
- Lama-3-8B-Instruct and Lama-3-Instruct-8B: please use uniform naming
- "(200 for ScienceWord and Webshop, 100 for textCraft)" -> 200 simulations
- in Table 1 "Critic Model (Ours)" -> CGI (Ours)
- Figure 4 should not appear before Fig. 3.
- caption Fig.3 "Ration" -> Ratio
- in references, Agentrm -> AgentRM (use "{AgentRM}" and check all papers)

**Ethical Concerns:**

["NO or VERY MINOR ethics concerns only"]

**Final Justification:**

The rebuttal process helped a lot clarifying what the authors are truly doing and why their method improves over just fine-tuning with an expert model. I now believe that the work is good, but I see a risk that the authors do not leverage the clarification discussion to write a much clearer paper. Hence my borderline accept.

**Limitations:**

Yes

**Paper Formatting Concerns:**

No concern

**Quality:**

2

**Strengths And Weaknesses:**

Strengths:
- the method seems to work well in practice, outperforming a large number of baselines in 3 representative environments
- the extra computational cost seems counter-balanced by the significant gain in performance

Weaknesses:
- some very related work is missing (see below)
- a key ablation is missing (see below)
- several points in the methods and empirical study are questionable (see below)

---

> ### Author Rebuttal · Authors · 2025-07-31
>
> > **Weakness 1:** The proposed framework is very close to SELU [1], apart from the fact that SELU addresses multimodal LLMs rather than purely text-based ones. In SELU they also have a critic-guided improvement mechanism.
> Thank you for pointing out this related work. We would like to clarify the key differences between our approach and SELU[1]:
>
> Thank you for bringing this concurrent work to our attention. While there are high-level similarities, CGI and SELU differ in fundamental ways:
>
> - **Feedback Type:** SELU's critic provides **trajectory-level binary feedback** (`{yes, no}`). In contrast, CGI's critic provides **step-level, rich, structured linguistic feedback** (grading, feasibility, revision suggestions). This is a core design difference.
> - **Critic Objective:** SELU's critic is a simple classifier to filter trajectories for training data. Our critic is a generative model trained to produce **actionable guidance**.
> - **Problem Setting:** SELU is designed for settings without expert data and works by simplifying tasks when the agent fails. Our work focuses on demonstrating that high-quality language feedback itself is a powerful mechanism for improving performance on complex, fixed tasks.
>
> We will add a detailed discussion of SELU to our related work section.
>
> > **Weakness 2:** A key ablation is missing: removing the initialization with expert trajectories.
>
> This is a critical point about the reliance on expert data. We have conducted the requested ablation study by replacing AgentGym's gold paths with GPT-4o generated trajectories on ScienceWorld, following the exact pipeline described in Section 3.2 to construct $D_{critique}$ for training the critic model.
>
> | Critic Training Data | Performance |
> | --- |:---:|
> | Llama3-8B (baseline) | 14.48 |
> | GPT-4o (for comparison) | 46.91 |
> | **CGI w/ Critic (trained on GPT-4o paths)** | **56.93** |
> | CGI w/ Critic (trained on expert paths) | 68.51 |
>
> The results are clear: **CGI remains highly effective even without gold-standard expert data.** Our method still achieves a performance of 56.93, significantly outperforming both the baseline and the GPT-4o model that generated the initial data. This shows the robustness of our framework. We will add this to the appendix.
>
> >**Weakness 3:** To insist on this potential weakness, if the authors have a way to generate expert trajectories, why learn another agent that will perform the same trajectories? Is it just a matter of distilling into a smaller size agent?
>
> This is not just distillation. The expert trajectories from AgentGym [2] were collected via many expensive sampling runs with GPT-4o. Our framework achieves two things:
>
> 1. **Superior Performance:** Our final CGI agent (72.91 on ScienceWorld) **significantly outperforms the expert model (GPT-4o at 46.91)** used to generate the initial data.
> 2. **Efficiency:** Our framework uses much smaller models (Llama-3-8B), making it far more efficient at inference time than running a large, proprietary model like GPT-4o.
>
> > **Weakness 4:** The results presented in Tables 1 and 2 seem to result from a single run for each condition. Is there some variability in the results if one performs the same experiment several times?
>
> Our main results use greedy decoding (temperature=0), which is deterministic. However, to address your valid concern about stochasticity, we ran a **new experiment** on ScienceWorld and WebShop with temperature=1 for 3 separate runs.
>
> | Environment | Mean ± Std |
> | --- |:---:|
> | ScienceWorld | 78.17 ± 1.33 |
> | WebShop | 76.17 ± 1.82 |
>
> The standard deviation is very low, confirming that **our method is robust and its performance is stable across runs.** We will add this analysis to the appendix.
>
> **Weakness 5: Verbal vs. numerical feedback comparison**
> > The comparison between verbal feedback and numerical feedback is potentially very interesting but the proposed study does not adequately focus on this specific question, as it might also be the critic accuracy that is different between DGAP, Explicit RM and CGI, and not only the feedback modality.
>
> Thank you for this insightful comment. As mentioned in Appendix G.1, we ensured fair comparison by using the same number of gold paths for training CGI, DGAP, and ExplicitRM. Our goal is precisely to compare different critic capabilities, which is why we selected two major categories: numerical methods (DGAP, Explicit RM) and verbal methods (CGI, self-refine), while keeping the actor model fixed as Llama3-8B.
>
> > Expanding on the above point, isn't the point that CGI proposes (potentially not proposed by the actor) action refinements and the actor is fine-tuned on the difference to these actions, whereas in DGAP and Explicit RM, the actor will be fine-tuned on the difference to the best-ranked action among the ones proposed by the actor?
>
> We would like to clarify the process: For CGI, the actor samples 5 candidate actions, the critic provides critique for each action, and the actor takes the next step based on the critic's feedback. For numerical methods, we also sample 5 candidate actions, and the critic selects the best-ranked action for the next step. Both approaches are completely aligned in terms of sampling count and critic evaluation frequency.
>
>
> >**Other Points:**
> - **Training on Failed Trajectories:** This is an interesting direction. Our current design focuses on learning from success to ensure the quality of the critic's training data. Learning from failure is a complementary research problem, as seen in methods like SELU that modify the task itself. We believe our focus on high-quality critique generation is a valid and important contribution.
> - **Fine-tuning Original Model (`πθ`):** As shown in Appendix F.1, Table 4, we use three types of data to train the actor at each iteration: (1) correct paths (mixture of $D_{gold}$ and $D_{correct}$ from model-environment interaction), (2) $D_{refine}$ (critic-action pairs from model-environment interaction), and (3) $D_{general}$ (general datasets to maintain general capabilities). As model capability improves, $D_{correct}$ and $D_{refine}$ increase, while $D_{gold}$ and $D_{general}$ remain constant.
> - **Train/Test Split:** **Yes**, we strictly followed the official splits from AgentGym [2]. There is no data leakage.
> - **Confusing Sentence:** We will rephrase to: "Our CGI approach, using an 8B critic, significantly boosts the performance of various actor models (including Llama-3-8B, Llama-3-70B, and fine-tuned variants) far beyond what GPT-4o can achieve alone."
> - **Longer Trajectories:** We will rephrase the analysis in Section 6.1 to be about "trajectory length" rather than "difficulty" to be more precise.
> - **Related Work & Typos:** We will move the related work to the main body and fix all typos you kindly pointed out.
> We initially did this due to space constraints. In the final version, we will move the Related Work section to the main text.

---

> > ### Comment · Reviewer_v7iq · 2025-08-01
> > **HOW does the critic get better than GPT-4o ?**
> >
> > I have read all the reviews and all the rebuttals to these reviews from the authors, and my understanding and opinion about this work has improved, but there is still a point that remains mysterious to me.
> >
> > If the critic is trained using supervised learning from labels generated by GPT-4o on expert trajectories, how can it get better than GPT-4o at labelling trajectories? How can it develop superior spatial reasoning and goal-directedness? In their answer to Reviewer 3ZoY, the authors provide two convincing examples, but they do not explain HOW or WHY the critic becomes superior. So I second the point of Reviewer 3ZoY that there lacks an analysis on why the proposed critic model can significantly outperform GPT-4o...

---

> > > ### Author Response · Authors · 2025-08-03
> > >
> > > Thank you for reconsidering our paper. We appreciate the thoughtful feedback and would like to further clarify the reasoning behind our approach:
> > >
> > > 1. **We use expert trajectories as references when synthesizing the training data for our critic model.** In this way, the critic model **combines the distilled knowledge present in expert trajectories, as well as GPT-4o’s capabilities** (Line 112-116, and Appendix D, Listing 2). Moreover, as mentioned in our response to Reviewer jS2Z, even when expert trajectories are replaced with sub-optimal ones (generated by GPT-4o), the resulting critic model still lead to measurable performance improvements.
> > > 2. As further clarified in our response to Reviewer jS2Z, all training data generated through this process is **filtered for correctness and format consistency.** This ensures that the critic model is trained on high-quality examples.
> > > 3. Our CGI framework includes an **iterative training process that continually improves the critic and actor's capabilities.** This synergy further contributes to the enhanced performance of the overall system.
> > >
> > > In summary, we believe it is reasonable to conclude that our critic model can outperform GPT-4o. We will incorporate the additional explanation into the revised version of the paper.

---

> > > > ### Comment · Reviewer_v7iq · 2025-08-03
> > > > **OK, but it could still be clearer**
> > > >
> > > > Though the authors' answer to my point is not that clear, it had the merit to draw my attention on the filtering mechanism that is described in Section 2, lines 64-72. I think this mechanism is crucial to explain how CGI can iteratively improve the trajectories it generates, in a way similar to reward-weighted regression (RWR, Peters et al., 2007) and Advantage-weighted regression (AWR, Peng et al. 2019).
> > > >
> > > > To me, this short piece of information should be moved into the methodology section (Section 3) and its importance should be better outlined.

---

> > > > > ### Author Response · Authors · 2025-08-03
> > > > >
> > > > > Thanks for the prompt reply. We are happy that our response has helped you better understand our approach. We will take your suggestion and revise our manuscript carefully to highlight how the data filtering mechanism works and emphasize its importance in ensuring the performance of our approach.
> > > > >
> > > > > Given that our responses have mostly addressed your questions, would you kindly reconsider the score? Thanks.

---

> > > > > > ### Comment · Reviewer_v7iq · 2025-08-04
> > > > > >
> > > > > > Don't worry, I will change my score positively, I'm just waiting for the discussion with the other reviewers to do so.

---

> ### Author Response · Authors · 2025-08-05
>
> Thank you for taking the time to review our paper! We are pleased that our response addressed your concerns. We also appreciate your engagement during the discussion.

---

> > ### Comment · Reviewer_v7iq · 2025-08-07
> > **Strong clarification, but a lot of rewrite required**
> >
> > I keep reading the authors' conversation with Reviewer 3ZoY. About comparing CGI with fine-tuning just with the expert model, the authors could also have answered what they answered me, the score by fine-tuning with GPT-4o being 46.91 and with CGI from GPT-4o paths being 56.93. The authors said that they will add this analysis to the appendix, I think it is quite crucial that they put it in the main paper.
> >
> > Finally, the last answer from the authors explaining that improvement takes place mostly during inference with the critic is a key explanation that has to be put much more forward.
> >
> > So, after this discussion, my understanding of the work and my opinion about it have significantly improved. My only remaining concern is whether the authors will be able to draw full benefit from this clarification process to write a much clearer paper. I'm afraid a lot of work is required.
> >
> > Given these last thoughts, I will just increase my score to borderline accept, leaving the final decision to accept or not to the AC.

---

> > > ### Author Response · Authors · 2025-08-07
> > >
> > > Thank you very much. We will take all your suggestions into account and revise our manuscirpt to make it more solid. We are grateful for the thorough discussions with you.

---

### Official Review · Reviewer_3ZoY · 2025-06-30

**Clarity:** 3
**Significance:** 3
**Originality:** 2
**Rating:** 4
**Confidence:** 4

**Summary:**

This work proposes Critique-Guided Improvement (CGI), a two-player framework that upgrades large-language-model agents with explicit, step-level natural-language feedback. An actor model first samples several candidate actions; a separately trained critic model then issues structured critiques that rate each action’s contribution, feasibility, and efficiency and includes concrete revision instructions. The actor is iteratively fine-tuned to incorporate these critiques, eliminating the brittleness of purely numerical rewards and the low quality of self-generated feedback. Using Llama-3-8B for both roles, CGI is evaluated on WebShop, ScienceWorld, and TextCraft. The 8-B critic outperforms GPT-4 as a feedback source, and after three refinement iterations the CGI agent achieves state-of-the-art results, surpassing strong closed- and open-source baselines and demonstrating especially large gains on long-horizon and hard tasks.

**Questions:**

Would the generated natural language feedback be still useful to powerful models that already have a good baseline performance?

**Ethical Concerns:**

["NO or VERY MINOR ethics concerns only"]

**Limitations:**

yes

**Quality:**

3

**Strengths And Weaknesses:**

Strength:
1. The paper focuses on critique-guided improvement, which is an important perspective to enhance the reflection capabilities of LLMs
2. The performance is great! Table 1 shows that the critic model outperforms the no-critique baseline by 40%+, and GPT-4o by 30%+.

Weakness:
1. In the essential, the idea of using feedback to improve LLMs is not entirely new.
2. There lacks analysis on why the proposed critic model can significantly outperform GPT-4o. Maybe two or three examples would make it more clear.

---

> ### Author Rebuttal · Authors · 2025-07-31
>
> Thank you for highlighting the importance of our work's perspective and its "great" performance.
>
> > **Weakness 1:** The idea of using feedback to improve LLMs is not entirely new.
>
> We agree that feedback-based improvement is an established direction. However, as we state in Lines 26-32, our novelty lies in **how we generate and utilize this feedback**. Prior work has struggled with two key challenges that our CGI framework is designed to solve:
>
> 1. **Low-Quality Feedback:** Relying on self-correction [2,3] or simple numerical rewards [1] provides limited guidance. Our key contribution is a specialized, trained critic that generates **high-quality, structured, and actionable natural language critiques.**
> 2. **Poor Utilization:** Agents often struggle to effectively integrate verbal feedback. Our second key contribution is the **iterative action refinement stage**, which explicitly fine-tunes the actor to understand and apply these critiques.
>
> Our work is the first to systematically address both the generation and utilization of high-quality language feedback in a unified framework.
>
> [1] Discriminator-guided embodied planning for LLM agent. ICLR 2025
>
> [2] Self-refine: Iterative refinement with self-feedback. NeurIPS 2024
>
> [3] Reflexion: Language agents with verbal reinforcement learning. NeurIPS 2023
>
>
> > **Weakness 2:** There lacks analysis on why the proposed critic model can significantly outperform GPT-4o.
>
> This is an excellent question. Our fine-tuned critic excels because it becomes a specialist for the task domain. Compared to the generalist GPT-4o, our critic develops superior **spatial reasoning** and **goal-directedness**. We will add the following examples to the appendix to illustrate this.
>
> - **Example 1 (Spatial Reasoning):**
>     - **State:** Agent is in the `workshop`. Task requires an object in the `bathroom`.
>     - **Action:** `focus on thermometer`
>     - **GPT-4o (Incorrect):** `Overall Grading: Excellent`. It approves the action based on the task description, ignoring the agent's location.
>     - **Our Critic (Correct):** `Overall Grading: Very Poor`. It correctly identifies the thermometer is not in the current room and suggests navigating first.
> - **Example 2 (Goal-Directedness):**
>     - **Task:** Find a living thing.
>     - **Action:** `move to the closet` (an exploratory but inefficient action).
>     - **GPT-4o (Inefficient):** `Overall Grading: Neutral`. It allows this exploratory step.
>     - **Our Critic (Efficient):** `Overall Grading: Poor`. It rejects the inefficient action and suggests a more direct one, like opening the door to explore other rooms.
>
> > **Q1:** Would the generated natural language feedback be still useful to powerful models that already have a good baseline performance?
>
> Thank you for pointing this out. To address this concern, we conducted experiments using powerful actor models (GPT-4o and Llama3-70B) guided by our critic model. The results are shown below:
>
> | Model | ScienceWorld | WebShop |
> |:-------|:--------------:|:---------:|
> | **Llama3-70B** | 49.20 | 8.35 |
> | **w/ Critic** | 72.44 | 52.20 |
> | **GPT-4o** | 46.91 | 25.48 |
> | **w/ Critic** | 75.91 | 58.59 |
>
> **Key findings:**
> - **Llama3-70B** shows substantial improvements: +23.24 points on ScienceWorld and +43.85 points on WebShop
> - **GPT-4o** also benefits significantly: +29.00 points on ScienceWorld and +33.11 points on WebShop
>
> These results demonstrate that even powerful models with strong baseline performance can benefit from our critic's guidance. We will add these results to Table 1 in the final version.

---

> > ### Author Response · Authors · 2025-08-04
> >
> > We appreciate your valuable feedback and thoughtful comments. Since it is near the end of the discussion period, could you kindly let us know if we have managed to address your concerns? Should you have any further questions, kindly let us know. We are more than happy to have further discussions with you. Thanks

---

> > > ### Comment · Reviewer_3ZoY · 2025-08-06
> > > **Thank you**
> > >
> > > Thanks a lot for the author's clarification!
> > >
> > > I agree with the author that one difference is that the trained critic model provides more fine-grained feedback. However, the iterative action refinement seems to be a known approach that is the way how we use the feedback. Additionally, I notice that  the critic model is finetuned with expert trajectory (otherwise it may not be able to give fine-grained feedback). In this case, I am wondering why not just use the expert data to finetune the action model, which seems to be more straightforward. Last but not least, since the critic model is finetuned with expert data, the numbers that use critic model may be considered as involving expert information, while other (experiment) numbers may not use it, which I think may not be very fair comparison.
> > >
> > > Sorry that it may be a bit late, so further experiments may not be necessary, but I hope that the authors can make their points more clear. Thanks!

---

> > > > ### Author Response · Authors · 2025-08-06
> > > >
> > > > Thank you for taking the time to review our paper. We sincerely appreciate your thoughtful engagement during the discussion phase. Below, we address the two concerns you raised:
> > > >
> > > > > **Q1: Why not directly use expert data to fine-tune the actor model?**
> > > >
> > > > As shown in **Table 1** of our paper, the combination of our trained critic model with the vanilla LLaMA3-8B outperforms the SFT-only LLaMA3-8B model by **+9.78%**. This demonstrates that **verbal feedback provides richer, more actionable guidance** compared to simple supervised learning from expert trajectories.
> > > > As mentioned in **Appendix Table 4**, we used the **same number of expert trajectories** for both training the critic model and fine-tuning the actor model across all three environments:
> > > > * **ScienceWorld**: 378
> > > > * **Webshop**: 427
> > > > * **TextCraft**: 441
> > > >
> > > > Furthermore, we observed that models trained via SFT tend to **underutilize critiques**, which motivated the **action refinement** stage to improve the model’s ability to leverage critique signals effectively. We will incorporate this analysis into **main results** in the final version of the paper for clarity.
> > > >
> > > > > **Q2: Concerns about fairness in comparison.**
> > > >
> > > > As detailed in **Appendix G.1 (Lines 859–960)**, all training methods in our comparison utilize the **exact same expert trajectories** (See Q1), ensuring completely fair evaluation. The key difference lies in how this data is leveraged: our approach converts it into natural language critiques, while baselines (**DGAP** and **Explicit RM**) use it for numerical scoring. Specifically, **DGAP** is trained to predict the **similarity** between candidate and expert actions, while **Explicit RM** learns to estimate the **log-likelihood ratio** between them.
> > > > We will further clarify this point in **Appendix G** of the final version to prevent any misunderstanding regarding the fairness of the comparison.

---

> > > > > ### Comment · Reviewer_3ZoY · 2025-08-06
> > > > > **Thanks**
> > > > >
> > > > > If the same expert trajectory is used to finetune both actor and critic model, then why the critic model can provide better action (e.g., open the door in the example 2 above) than the action model? Is this because finetuning action model does not use critic generated by experts (GPT-4 mentioned in line 111)?  Thanks!

---

> > > > > > ### Author Response · Authors · 2025-08-06
> > > > > >
> > > > > > Thank you for your insightful question. The superior performance of the critic model stems from fundamental differences in the **inference mechanism**.
> > > > > >
> > > > > > During **inference with the critic model** (Section 3.1, Lines 85-89), the process involves two distinct phases:
> > > > > > 1. **Candidate Generation**: The actor generates M candidate actions
> > > > > > 2. **Selection and Refinement**: The critic evaluates each candidate, providing structured feedback (Contribution, Feasibility, Efficiency, overall grade, and revision suggestions)
> > > > > >
> > > > > > The key insight is that selection/evaluation is **easier than generation from scratch.** The critic either selects the best candidate when reasonable options exist, or provides revision suggestions to guide the actor beyond suboptimal candidates. This significantly reduces the difficulty compared to directly generating optimal actions.
> > > > > >
> > > > > > In contrast, **when using only the actor model during inference**, there is no evaluation-refinement loop, the model must generate the optimal action directly.
> > > > > >
> > > > > > This architectural difference explains why the critic can achieve better performance even when both models are trained on identical expert trajectories. We hope this clarification addresses your concern. Thank you again for your valuable feedback.

---

> ### Author Response · Authors · 2025-08-08
>
> Dear reviewer, since the discussion deadline is approaching, could you kindly let us know if our responses have addressed your concerns? We appreciate your time and feedback.

---

### Official Review · Reviewer_jS2Z · 2025-07-01

**Clarity:** 2
**Significance:** 3
**Originality:** 2
**Rating:** 3
**Confidence:** 4

**Summary:**

This paper introduces Critique-Guided Improvement (CGI), a novel two-player framework designed to enhance the performance of large language model (LLM)-based agents. The framework consists of an actor model that interacts with an environment and a critic model that generates detailed, natural language feedback. In the Critique Generation stage, the critic model provides precise evaluations and actionable suggestions. In the Action Refinement stage, the actor model learns to integrate this feedback through iterative fine-tuning, improving both reasoning and decision-making capabilities.

**Questions:**

Please refer to the weaknesses section. Specifically: 1. It is suggested that the authors provide a detailed explanation of how the environmental reward (R) is calculated, particularly whether it only considers task completion or also includes factors such as action efficiency, path optimization, etc. (Is there a critique-based reward mechanism?). 2. Could the authors clarify the model’s handling mechanism when encountering incorrect candidate actions, especially how it backtracks and corrects erroneous actions? 3. The numerical definition of long-term tasks and ablation studies on action refinement should also be added in the experimental section to help further validate the model's performance across different tasks and iterations.

**Ethical Concerns:**

["NO or VERY MINOR ethics concerns only"]

**Final Justification:**

I have read the rebuttal and the comments of the other reviewers. My final rating is borderline reject.

**Limitations:**

Although the paper demonstrates the effectiveness of the CGI method, it lacks a discussion of its limitations or potential issues in real-world applications. For example, can the CGI method effectively handle extreme situations or environments with highly complex tasks? What about the model's scalability and computational complexity?

**Paper Formatting Concerns:**

There are no major formatting issues with the paper.

**Quality:**

3

**Strengths And Weaknesses:**

Strengths

The paper introduces a novel Critique-Guided Improvement (CGI) method, which significantly enhances the model's performance in complex interactive tasks by combining a critique model with action refinement. The structure of the paper is clear, logically rigorous, and easy to understand. Each section is thoroughly developed, and the transitions between sections are smooth, improving the overall readability of the paper.

Weaknesses

1、Ambiguity in the Definition of Environmental Reward: The paper does not clearly explain the specific calculation of environmental reward (R), particularly whether the reward is solely based on task completion or if it also considers factors such as action efficiency, path optimization, and other aspects.

2、Lack of a Structured Reward Function: In Section 3.2, the paper discusses the structure of the critique model and feedback generation, but it does not further address how to utilize a structured reward function to optimize and standardize the critique generation process. If a format-based reward function were introduced, would the generated feedback be more standardized? It is suggested to provide further clarification on this.

3、Inadequate Handling of Incorrect Candidate Actions: The paper focuses on how to improve the critique model's matching of candidate actions with ideal "annotated answers," but it does not address how the model should handle cases where no accurate candidate actions are generated. How should the model deal with incorrect actions? How should backtracking corrections be applied? It is suggested to provide more clarification on this aspect.

4、In the experimental section, although the paper mentions that task difficulty is related to trajectory length, it does not provide a clear numerical threshold or boundary to define long-term and short-term tasks.

5、While action refinement is shown to improve the model's performance, there is a lack of clear ablation comparison experiments. It is recommended to include relevant explanations or experiments.

---

> ### Author Rebuttal · Authors · 2025-07-31
>
> > **Weakness 1:** Ambiguity in the Definition of Environmental Reward (R).
>
> Thank you for asking for clarification. The environmental reward `R` is defined **solely by task completion**, following the standard setup of the AgentGym benchmark [1] we use.
>
> - **ScienceWorld**: A score from 0 to 100 based on sub-task completion.
> - **TextCraft & WebShop**: A binary reward (1 for success, 0 for failure).
>
> Our reward function does not include factors like efficiency or path length. This is a deliberate choice to ensure a fair and direct comparison with prior work on these standard benchmarks. We will add this explicit definition to Section 4.1.
>
> [1] AgentGym: Evolving Large Language Model-based Agents across Diverse Environments
>
> > **Weaknesses 2&3:** Lack of a structured reward function and inadequate handling of incorrect candidate actions.
>
> We respectfully clarify that our framework explicitly handles both of these points.
>
> 1. **Structured Critiques:** As described in Section 3.2, our critiques are **highly structured**. The critic is trained to generate feedback across five dimensions: **Contribution, Feasibility, Efficiency, Overall Grading, and a Suggested Revision**. We enforce this structure by filtering the training data, ensuring the critic learns to produce standardized and effective feedback.
> 2. **Handling Incorrect Actions:** Our framework is designed for the common scenario where **all candidate actions might be incorrect.** The critic's role is not just to pick the best action, but to provide a `Suggested Revision` that guides the actor toward a better, potentially entirely new action.
>
> To make this concrete, we will add the following example to the appendix, where all initial actions are flawed:
>
> - **Task:** Use a thermometer in the bathroom.
> - **State:** The agent is in the kitchen.
> - **Bad Candidate Action:** `focus on thermometer` (it's in another room).
> - **Our Critic's Feedback:** `Overall Grading: Very Poor. Suggested Revision: The agent should first 'open door to hallway' to begin navigating toward the bathroom...`
>
> This shows how CGI provides actionable guidance even when the initial proposals are unworkable.
>
>
> >**Weakness 4** In the experimental section, although the paper mentions that task difficulty is related to trajectory length, it does not provide a clear numerical threshold or boundary to define long-term and short-term tasks.
>
> Thank you for requesting this clarification. In our experiments, task length classification is based directly on the **ScienceWorld** environment [1], which provides gold-standard trajectories from 30 hand-coded oracle agents. Following prior work [2, 3], we adopt the established categorization:
>
> | Task Type | *Len |
> |-----------|---------------------------|
> | **Easy** |              11.76 |
> | **Medium** |         28.58 |
> | **Hard** |              94.30 |
>
> These lengths correspond to the **average number of steps** taken by the oracle agent for each task difficulty level. This classification enables us to analyze how CGI performs across varying task complexities, as shown in Figure 4. We will include this detailed information in Section 4.1 and ensure consistent terminology throughout the paper.
>
> [1] ScienceWorld: Is Your Agent Smarter than a 5th Grader? *EMNLP 2022*
>
> [2] SWIFTSAGE: A Generative Agent with Fast and Slow Thinking for Complex Interactive Tasks. *NeurIPS 2023*
>
> [3] Discriminator-Guided Embodied Planning for LLM Agent. *ICLR 2025*
>
>
> > **Weakness 5:** While action refinement is shown to improve the model's performance, there is a lack of clear ablation comparison experiments. It is recommended to include relevant explanations or experiments.
>
> Thank you for raising this concern. We want to clarify that we have conducted comprehensive ablation studies, which are presented in the appendix:
>
> **Ablation Study 1 - Data Component Analysis (Appendix H)**: We conducted an ablation study on the contribution of three types of data (`D_correct`, `D_refine`, and `D_general`) used to enhance the actor model during action refinement. Our results confirm that critique-action pairs (`D_refine`) provide the most significant contribution to performance improvement.
>
> **Ablation Study 2 - Iteration Count Analysis (Appendix F.3)**: We analyzed the effect of different numbers of action refinement iterations, demonstrating that three iterations achieve optimal performance while a fourth iteration provides no additional benefit.
>
> We believe these ablation studies provide sufficient evidence for the effectiveness of our approach. If you have specific aspects you would like us to investigate further, we would be happy to include additional ablation experiments.
>
> **Limitations**
>
> > Although the paper demonstrates the effectiveness of the CGI method, it lacks a discussion of its limitations or potential issues in real-world applications.
>
> Thank you for this point. We analyze computational cost in Appendix E.2, showing that CGI provides a +43.31% performance gain for a ~4x inference time cost, a much more favorable trade-off than using a larger model like GPT-4o (+4.29% gain for a 7x time cost). While we selected complex, long-horizon environments to approximate real-world challenges, we agree that scaling to fully open-ended tasks is a key direction for future work. We will add a dedicated Limitations section to discuss this.

---

> ### Author Response · Authors · 2025-08-05
>
> Sincerely thank you for taking the time to review our paper. We are glad to know that our response addressed your concerns. Thanks for your engagement during the discussion!

---

### Official Review · Reviewer_5w5J · 2025-07-02

**Clarity:** 2
**Significance:** 3
**Originality:** 4
**Rating:** 5
**Confidence:** 4

**Summary:**

This paper propose CGI a two-player framework consisting of an actor module, which explores and takes actions in an environment, and a critic module, which provides natural language critique that provides action quality judgement and feedback for revision. The critique is distilled from a larger model GPT4 by selecting trajectories containing actions and critiques leading to successful outcome. The actor seems to be trained by jointly optimizing the probability of the policy taking improved actions given observations and revising proposed actions incorporating the critique. Results showed that CGI robustly outperforms other critique methods and iterative-SFT-based method alone in three different tasks (ScienceWorld, WebShop, TextCraft). Further analysis suggest that the critic usually takes effect early on during actors' explorations.

**Questions:**

- Does table 1 result come from the 3rd iteration as Table 2 and 4?
- $L_{actor}$ term is quite unclear, and does not track with the previous text. Please fix it so it's more clear.
- minor suggestion: change “critique” to CGI to avoid confusion in FIgure 5.

**Ethical Concerns:**

["NO or VERY MINOR ethics concerns only"]

**Final Justification:**

The original draft proposed a novel idea with strong results, although the writing in certain sections, particularly in method section, are not clear. The confusion I had regarding to the draft is clarified during the rebuttal, hence, I am raising my score.

**Limitations:**

No, the author did not talk about any limitations of this work and i have mentioned two in the Weaknesses.

**Quality:**

3

**Strengths And Weaknesses:**

Strengths:
- As the field is moving towards building more agentic capabilities in open-ended domains where verification is non-trivial, this work provide a way to improve LLMs' ability to provide and improve upon natural language (NL) feedback is timely.
- The results are very solid. Comparison with bases for both critic and actor components in three non-trivial domains are provided, and the results showed the two-player framework led to significant and robust improvements across different models.
- I appreciate the further analysis section provided in section 6, it is meaningful to see how and where the most of the gains comes from. Also interesting to see a mostly positive scaling of performance w.r.t number of candidate actions. Although seems to decrease a bit after 7.

Weaknesses:
- while I find most of the paper clear, I have a hard time following the method section, particularly the part describing how the actor is trained. Per state of this paper, there is not enough info to reimplement the method. Here are a list of my confusions:
  - there is multiple versions of datasets D mentioned, but not all of them are clearly defined (D_general) whereas some annotations are confusing D_critique (but it's from the expert)
  - $L_{actor}$ formula does not reflect description btw L132 and L143, and also $y$ and $\beta$ are not defined anywhere in text
- this general direction is important but there's a few weaknesses in CGI:
  - the existence of expert critic is required, which limits the ceiling of the critic performance, and also will not apply to domains where cheap expert critic is hard to find,
  - the critic stays static during the actor improvement phase, while realistically it should evolve alongside the actor policy changes. However, the current implementation prevents that.

---

> ### Author Rebuttal · Authors · 2025-07-31
>
> Thank you for the positive feedback on the timeliness of our work and the robustness of our results. We address the concerns about clarity and the critic model below.
>
> >  **Weakness 1:** Method clarity, undefined symbols ($D_{general}$, $D_{critique}$, $y$, $\beta$), and unclear actor loss $L_{actor}$.
>
> Thank you for pointing out the lack of clarity. We will add a comprehensive symbol table to the appendix and clarify all definitions in the main text.
>
> **Dataset Definitions**
>
> | Symbol |Meaning|
> |---------|---------|
> | $\mathcal{D}_{\text{critique}}$ | Expert critique dataset generated by expert critic (GPT-4o), used for training the critic model as described in Section 3.2 |
> | $\mathcal{D}_{\text{correct}}$ | Correct trajectory dataset collected during each iteration of the action refinement process |
> | $\mathcal{D}_{\text{refine}}$ | Critique-action pairs dataset used for learning how to utilize critiques effectively |
> | $\mathcal{D}_{\text{general}}$ | General conversational dataset (e.g., ShareGPT) used to maintain general language modeling capabilities during fine-tuning |
>
> **Loss Function Parameters**
>
> | Symbol | Meaning |
> |---------|---------|
> | $\mathcal{L}_{\text{actor}}$ | Actor model loss function with three components: learning from expert/correct trajectories, learning to utilize critiques, and maintaining general capabilities |
> | $\beta$ | Weight parameter balancing agentic task learning vs. general capabilities (typically β = 0.8) |
> | $y$ | General text responses from datasets like ShareGPT for maintaining language modeling abilities |
>
> We will revise Lines 132-143 to ensure the mathematical formulation aligns with the textual description and add these definitions explicitly in the main text.
>
>
> > **Weakness 2:** The expert critic requirement limits the performance ceiling, and the critic is static.
>
> This is an excellent point. Our framework has two key strengths here:
>
> 1. **The expert does NOT create a performance ceiling.** As shown in Table 1, our fine-tuned Llama-3-8B critic **already surpasses the GPT-4o expert by a large margin (+29.16%)**. This demonstrates that our method distills and refines the expert's knowledge, rather than being capped by it.
> 2. **The critic can be dynamically updated.** Following your insightful suggestion, we ran a **new experiment** with an alternating critic-actor update scheme. We used the first iteration's critic to generate new critique data and retrained the critic.
>
> **Results on ScienceWorld and TextCraft:**
>
> | Method | ScienceWorld | TextCraft |
> |:--------|:--------------:|:-----------:|
> | Llama-3-8B (baseline) | 14.48 | 10.00 |
> | w/ Static Critic | 72.91 | 62.00 |
> | w/ Updated Critic | 74.56 | 65.00 |
>
> As the results show, an updated critic yields further performance gains. This confirms the critic is not limited to being static and can evolve with the actor. We will add these results and the methodology to the final paper.
>
> **Questions:**
>
> > Does Table 1 result come from the 3rd iteration as Table 2 and 4?
>
> **No.** Table 1 is designed to isolate and evaluate the quality of the critic model itself. It shows the performance of a base Llama-3-8B actor guided by different critics, without any iterative actor refinement. This is in contrast to Tables 2 and 4, which show the full CGI framework with iterative actor improvement. We will state this explicitly in the caption of Table 1 to make this distinction clear.
> > Minor suggestion: change "critique" to CGI to avoid confusion in Figure 5.
>
> Thank you for the suggestion. We agree this improves clarity and will update Figure 5 in the revised version.

---

> > ### Author Response · Authors · 2025-08-04
> >
> > Thanks for your effort in providing us such insightful comments. Since it is near the end of the discussion period, could you kindly let us know if we have managed to address your concerns? Should you have any further questions, kindly let us know. We are more than happy to have further discussions with you. Thanks

---

> ### Comment · Reviewer_5w5J · 2025-08-07
>
> Thank you for your thoughtful and thorough response. You have addressed my confusion of the paper, and I believe with proposed changes made to the draft, it would be a very good paper and Im looking forward to the final version. I will improve my score.

---

> > ### Author Response · Authors · 2025-08-07
> >
> > Sincerely thank you for taking the time to review our paper. We are glad to know that our response addressed your concerns. Thanks for your engagement during the discussion!

---

### Note · Authors · 2025-08-12

We are grateful to all reviewers for their valuable insights and constructive feedback.

In the rebuttal phase, we have addressed the following concerns:
1. **Method Novelty** (*Reviewers 3ZoY, v7iq*): Our work tackles both the generation and utilization of high-quality language feedback within a single framework. We enhance feedback quality through specialized critic training and improve utilization via iterative action refinement.

2. **Why Our Trained Critic Outperforms GPT-4o** (*Reviewers 5w5J, 3ZoY, v7iq*): Our fine-tuned critic excels by **integrating expert knowledge with GPT-4o capabilities**, filtered for correctness and format consistency. It surpasses GPT-4o in **spatial reasoning and goal-directedness** within the task domain.

3. **Concerns About Performance Claims** (*Reviewers 3ZoY, v7iq*):  Our two-phase inference uses: (1) actor generates candidate actions, (2) critic selects and refines using structured feedback. **Evaluation and refinement is easier than generating optimal actions from scratch**, which explains our superior performance even with identical expert trajectories. Our ablation study shows our critic model remains effective when trained with sub-optimal GPT-4o trajectories.

We are pleased that all reviewers expressed satisfaction with our responses. Reviewer 5w5J confirmed our responses addressed their concerns and that the proposed changes would make this *"a very good paper"*, while Reviewer v7iq noted their *"significantly improved understanding and opinion"* of our work. We will incorporate all suggestions raised during the discussion period into the final version. **All experimental results mentioned in our rebuttal are prepared and ready for integration into the final version.**

We sincerely appreciate the reviewers' constructive engagement, which has improved both the technical rigor and clarity of our paper.

---

### Decision · Program_Chairs · 2025-09-17

**Decision:**

Accept (poster)

**Comment:**

This paper introduces Critique-Guided Improvement (CGI), a two-player actor–critic framework where the critic provides structured natural language feedback and the actor is iteratively fine-tuned to incorporate it. Evaluated on ScienceWorld, WebShop, and TextCraft, CGI achieves strong gains, even surpassing GPT-4o as a feedback source.

Strengths: The work tackles an important direction for LLM agents, and demonstrates convincing improvements across three benchmarks. The use of structured critiques and iterative refinement is well-motivated and clearly impactful. Reviewers appreciated the thorough ablations, scaling analyses, and additional experiments provided in rebuttal.

Weaknesses: The main concern across reviews is clarity. Several features (e.g., ablations, structured critique design, explanation for critic > GPT-4o) were already in the submission but buried in the appendix or insufficiently emphasized. This led to confusion about novelty and completeness. Reviewers also noted overlap with related work (e.g., SELU), though rebuttal clarifications distinguished CGI’s contributions.

After rebuttal, most concerns were resolved, and residual issues are presentation-related.

Final Recommendation: Accept. The work is technically solid and impactful, but the final version must integrate rebuttal clarifications and highlight key details (datasets, ablations, inference-time benefits of the critic) in the main text for clarity and reproducibility.